# Proximal and Remote Sensing Data Integration to Assess Spatial Soil Heterogeneity in Wild Blueberry Fields

**Allegra Johnston** [1,2], **Viacheslav Adamchuk** [2] , **Athyna N. Cambouris** [1,*], **Jean Lafond** [3], **Isabelle Perron** [1], **Julie Lajeunesse** [3], **Marc Duchemin** [1] **and Asim Biswas** [4]

1. Quebec Research and Development Centre, Agriculture and Agri-Food Canada, 2560 Hochelaga, Québec City, QC G1V 2J3, Canada
2. Department of Bioresource Engineering, McGill University, 21111 Lakeshore Road, Ste-Anne-de-Bellevue, QC H9X 3V9, Canada
3. Normandin Research Farm, Agriculture and Agri-Food Canada, 1468 St-Cyrille Street, Normandin, QC G8M 4K3, Canada
4. School of Environmental Sciences, University of Guelph, 50 Stone Road East, Guelph, ON N1G 2W1, Canada
* Correspondence: athyna.cambouris@agr.gc.ca

**Abstract:** Wild blueberries (*Vaccinium angustifolium* Ait.) are often cultivated uniformly despite significant within-field variations in topography and crop density. This study was conducted to relate apparent soil electrical conductivity ($EC_a$), topographic attributes, and multi-spectral satellite imagery to fruit yield and soil attributes and evaluate the potential of site-specific management (SSM) of nutrients. Elevation and $EC_a$ at multiple depths were collected from two experimental fields (referred as $Field_{Und}$, $Field_{Flat}$) in Normandin, Quebec, Canada. Soil samples were collected at two depths (0–0.05 m and 0.05–0.15 m) and analyzed for a range of soil properties. Statistical analyses of fruit yield, soil, and sensor data were used to characterize within-field variability. Fruit yield showed large variability in both fields ($CV_{Und}$ = 54.4%, $CV_{Flat}$ = 56.5%), but no spatial dependence. However, several soil attributes showed considerable variability and moderate to strong spatial dependence. Elevation and the shallowest depths of both the Veris (0.3 m) and DUALEM (0.54 m) $EC_a$ sensors showed moderate to strong spatial dependence and correlated significantly to most soil properties in both study sites, indicating the feasibility of SSM. In place of management zone delineation, a quadrant analysis of the shallowest $EC_a$ depth vs. elevation provided four sensor combinations (scenarios) for theoretical field conditions. ANOVA and Tukey–Kramer's post hoc test showed that the greatest differentiation of soil properties in both fields occurred between the combinations of high $EC_a$/low elevation versus low $EC_a$/high elevation. Vegetation indices (VIs) obtained from satellite data showed promise as a biomass indicator, and bare spots classified with satellite imagery in $Field_{Und}$ revealed significantly distinct soil properties. Combining proximal and multispectral data predicted within-field variations of yield-determining soil properties and offered three theoretical scenarios (high $EC_a$/low elevation; low $EC_a$/high elevation; bare spots) on which to base SSM. Future studies should investigate crop response to fertilization between the identified scenarios.

**Keywords:** precision horticulture; proximal soil sensors; apparent soil electrical conductivity; SPOT satellite image; management zone; sensor combinations



## 1. Introduction

Wild blueberries (*Vaccinium angustifolium* Ait.), are a leading Canadian fruit export worth an estimated $239 M and distributed in more than 30 countries [1]. From 2011–2019, wild blueberries contributed an average annual farmgate value of $98.4 M in Canada [2]. Due to their winter-hardiness and ability to thrive in naturally acidic, sandy soils, wild blueberries make up a significant portion of the agricultural industries of Northern New England, Atlantic Canada, and Quebec.

Because of spatially heterogeneous growing conditions—including topography, water availability, and crop density—wild blueberries could benefit from site-specific nutrient management (SSM). Some studies have investigated SSM in wild blueberry production. Variable rate fertilization of wild blueberry based on proximal topographic data (slope) has demonstrated increased efficiency in nutrient application and reduced subsurface water contamination [3]. Findings by Farooque et al. (2012) found significant differences among management zones (MZs) based on yield and soil properties, and furthermore found significant positive correlations of electrical conductivity ($EC_a$) with soil attributes and fruit yield, suggesting the feasibility of using $EC_a$ data to delineate MZs [4].

Site-specific management based on temporally stable soil attributes are preferred to yield-based SSM when high yield variability is observed [5]. As a biennial crop, wild blueberry is susceptible to variability due to variations in seasons, growing conditions, and management history. Studies have found that soil apparent electrical conductivity ($EC_a$) relates to several yield-determining properties such as soil organic matter, moisture content, and soil texture [6–9]. In unsaturated, non-saline soils, $EC_a$ has been found to reflect variations in both moisture availability and soil texture [8]. $EC_a$ may be measured via time domain reflectometry (TDR), electromagnetic induction (EMI), or electrical resistivity. TDR is a slower, point-based measurement solution which is difficult to adapt to field scale [10]. Alternatively, EMI and resistivity $EC_a$ sensors provide quick, on-to-go proximal sensing at a sampling density that can detect local-scale variability [11]. When paired with a global navigation satellite system (GNSS) receiver, proximal $EC_a$ measurements are georeferenced.

In parallel, mapping crop density and delineating bare spots is a field of increasing interest for wild blueberry and precision agriculture, given the prevalence of bare spots in young and mismanaged fields [3,12,13]. One study in Nova Scotia reported the percentage of bare spots in wild blueberry study sites varied between 30–50% [13]. Excessive fertilization of bare spots may be economically inefficient and risks contaminating water. Given their nonlinear response to nutrient application, it is recommended that bare spots be managed separately [14]. In their study of MZ delineation, Farooque et al. (2012) suggested defining bare spots as a separate class while delineating MZs for nutrient input savings [4]. Previous research has mapped bare spots with a Global Navigation Satellite System (GNSS) [3,15] or digital color photography [12,13]. As an alternative, satellite imagery provides non-invasive and inexpensive data of large sites which can be rapidly analyzed and classified with vegetation indices (VI). MZ delineation based on VIs have been explored in grain crops [16,17], but the use of satellite imagery to delineate bare spots in wild blueberry fields is scarce.

Thus, a combination of proximal and remote sensing data provides improved information from their complementarity and is rarely addressed for blueberry production. Proximally sensed soil apparent electrical conductivity ($EC_a$) and topography provide quick, temporally stable, and dense data auxiliary to yield-determining properties. Soil $EC_a$ has been demonstrated to correlate with several soil properties including soil organic matter, nutrient availability, moisture content, and texture [6,9,18], while elevation and derived topographic information (e.g., topographic wetness index, slope, elevation) influence water holding capacity, nutrient accumulation, and water movement.

A common method of data separation for site-specific management is the employ of an unsupervised clustering algorithm such as fuzzy c-means to group similar data into MZs, thereby classifying similar values into contiguous zones for uniform treatment within zones [17,19,20]. The method is robust and widely used, but when several data layers are combined, clusters tend to reflect the data layer of greatest variability. Alternatively, the approach in this research aims to equally weigh $EC_a$ and topography data. A simple quadrant method was used to subset the study sites into four theoretical combinations (i.e., quadrants) according to their $EC_a$ and elevation values: $EC_{Low}Topo_{Low}$, $EC_{Low}Topo_{High}$, $EC_{High}Topo_{Low}$, $EC_{High}Topo_{High}$. These theoretical combinations identified extreme and

spatially continuous areas which represented unique agro-environmental conditions that may affect soil process and therefore crop yield.

The principal objectives of this study were: (1) to characterize within-field variation of wild blueberry crop growing conditions; (2) to determine if combining proximally sensed $EC_a$ and topographic data with remote sensing satellite imagery could significantly distinguish soil properties that may influence fruit yield within the study sites.

## 2. Materials and Methods

### 2.1. Study Sites

Two commercial fields, designated as $Field_{Flat}$ and $Field_{Und}$, were selected for the study (Figure A1). The fields were located 6 km southwest of Normandin, QC (48.8369° N, 72.5279° W) and north of the Ashuapmushuan River. Soil in the region is primarily podzolic, mixed with finer eolian deposits [21]. It is characterized by a rich organic surface layer followed by an eluviated mineral layer, and an illuviated layer where Aluminum and Iron redeposit. Podzols are characteristically acidic at the surface, and pH increases with depth. Drainage varies from moderate to good with topography ranging from flat to some undulation. $Field_{Flat}$ (11.3 ha) represented a uniform low-lying topography ranging from 123 m to 125 m elevation, and $Field_{Und}$ (13.2 ha) represented a more heterogeneous topography with elevation ranging from 127 m to 136 m.

### 2.2. Experimental Design and Field Methods

Soil and fruit yield samples were collected on 8–9 August 2016, in both fields with a 33 m × 33 m grid sampling scheme for a total of N = 136 points in $Field_{Und}$ and N = 116 points in $Field_{Flat}$. Blueberries were harvested with a hand-held rake from a square meter of blueberry bushes at each sample location. The weight of the fresh blueberries was measured and recorded on site. Intensive soil sampling was conducted post-harvest in October 2016 at the same locations from the organic horizon (0–0.05 m) and the mineral horizon (0.05–0.15 m). A composite of four soil cores was taken from a 1 m radius of the sample point to provide a representative sample.

### 2.3. Soil Analysis

Soil samples were air-dried, weighed, and ground to 2 mm for textural and laboratory chemical analysis. Total Carbon (C) and total Nitrogen (N) content were evaluated with the Elementar vario MAX CN analyzer (Elementar Analysensysteme GmbH, Hanau, Germany). A Mehlich-III soil extractant was used to extract Iron (Fe), Aluminum (Al), Phosphorus (P), Potassium (K), Calcium (Ca), and Magnesium (Mg) [22]. Soil P content was determined by colorimetry (Lachat Instruments, model 8500, series 2, Loveland, CO, USA) [23]. Soil K content was measured with flame emission spectrophotometry [24]. The soil Ca and Mg contents were determined with atomic absorption spectrophotometry (Agilent Technologies, model 200, series AA, Santa Clara, CA, USA). Soil pH was determined from water suspension (1:1, *v/v*) [25]. The P/Al ratio was calculated from the Mehlich-III extracted P and Al, as literature has shown it to be a useful indicator for P accumulation in Quebec mineral soils [26].

Soil particle size was determined for all soil samples at the 0.05–0.15 m depth (i.e., soil mineral horizon) using the pipette method [27]. Sand partitioning was examined given that percentage of silt and clay were expected to be low in podzolic soils. Texture was categorized in terms of grams per kilogram of very coarse sand (1.0 to 0.5 mm), coarse sand (0.5 to 0.25 mm), medium sand (0.25 to 0.10 mm), fine sand (0.1 mm to 0.05 mm), very fine sand (0.05 to 0.002 mm), total silt, and total clay according to the Canada Soil Survey Committee standards [28]. All descriptive statistics of soil analysis are presented in Table 1.

**Table 1.** Values of Mean, standard deviation from Mean (*SD*), and coefficient of variation (*CV*) for key soil properties at 0.00–0.05 m and 0.05–0.15 m depths, DUALEM horizontal co-planar (HCP), perpendicular co-planar (PRP) conductivities, Veris conductivities, and wild blueberry yield collected after the 2016 harvest from Field$_{Und}$ and Field$_{Flat}$ experimental fields in Normandin, Quebec.

| | Unit | Field$_{Und}$ | | | Field$_{Flat}$ | | |
|---|---|---|---|---|---|---|---|
| | | Mean | SD | CV | Mean | SD | CV |
| Soil attributes 0–0.05 m depth | | | | | | | |
| Total Nitrogen (N) | % | 0.46 | 0.270 | 59.1 | 0.44 | 0.25 | 56.7 |
| Total Carbon (C) | % | 11.1 | 6.50 | 58.7 | 8.80 | 5.10 | 57.5 |
| Soil pH$_{water}$ | – | 4.70 | 0.50 | 10.6 | 4.50 | 0.40 | 7.80 |
| Phosphorous (P) | mg kg$^{-1}$ | 63.0 | 54.0 | 85.1 | 39.0 | 48.0 | 124.0 |
| Potassium (K) | mg kg$^{-1}$ | 107 | 70.1 | 65.3 | 93.0 | 56.0 | 60.4 |
| Calcium (Ca) | mg kg$^{-1}$ | 361 | 76.0 | 21.2 | 387 | 99.0 | 25.6 |
| Magnesium (Mg) | mg kg$^{-1}$ | 107 | 71.9 | 67.0 | 78.0 | 53.0 | 68.7 |
| Aluminum (Al) | mg kg$^{-1}$ | 889 | 287 | 32.3 | 939 | 294 | 31.3 |
| Iron (Fe) | mg kg$^{-1}$ | 1502 | 933 | 62.1 | 465 | 338 | 72.7 |
| P/Al ratio | – | 0.039 | 0.038 | 97.6 | 0.069 | 0.049 | 71.3 |
| Soil attributes 0.05–0.15 m depth | | | | | | | |
| Total Clay | g kg$^{-1}$ | 23.5 | 5.20 | 22.1 | 26.5 | 6.10 | 23.1 |
| Total Silt | g kg$^{-1}$ | 119.7 | 75.6 | 63.1 | 77.5 | 30.5 | 39.3 |
| Total Sand | g kg$^{-1}$ | 857 | 74.0 | 8.60 | 896 | 30.0 | 3.4 |
| Very coarse sand [1] | g kg$^{-1}$ | 12.0 | 13.7 | 113.7 | 25.4 | 15.3 | 60.0 |
| Coarse sand [2] | g kg$^{-1}$ | 99.9 | 88.8 | 88.9 | 170 | 88.9 | 52.3 |
| Medium sand [3] | g kg$^{-1}$ | 284.8 | 163.2 | 57.3 | 357 | 103 | 28.9 |
| Fine sand [4] | g kg$^{-1}$ | 312.2 | 123.2 | 39.5 | 280 | 126 | 45.0 |
| Very fine sand [5] | g kg$^{-1}$ | 147.8 | 130.3 | 88.1 | 63.3 | 49.3 | 77.9 |
| Fruit yield and Sensor data | | | | | | | |
| Fruit yield | g m$^{-2}$ | 643 | 350 | 54.4 | 399 | 225 | 56.5 |
| HCP 1.0 [6] | mS m$^{-1}$ | 4.29 | 0.73 | 17.0 | 4.26 | 0.35 | 8.20 |
| PRP 1.1 [7] | mS m$^{-1}$ | 1.33 | 0.14 | 10.7 | 1.02 | 0.11 | 10.5 |
| HCP 2.0 [8] | mS m$^{-1}$ | 3.84 | 0.31 | 8.10 | 2.95 | 0.22 | 7.60 |
| PRP 2.1 [9] | mS m$^{-1}$ | 1.65 | 0.11 | 6.90 | 1.31 | 0.11 | 8.60 |
| Veris Shallow [10] | mS m$^{-1}$ | 3.21 | 0.10 | 2.40 | 2.70 | 0.10 | 2.30 |
| Veris Deep [11] | mS m$^{-1}$ | 2.86 | 0.60 | 22.0 | 2.30 | 0.80 | 34.2 |
| Elevation | m | 132.2 | 2.60 | 1.90 | 124.3 | 0.50 | 0.40 |
| Slope | deg | 1.90 | 2.60 | 134 | 0.90 | 1.20 | 130 |
| TWI [12] | – | 6.40 | 3.30 | 51.6 | 5.00 | 2.80 | 56.4 |

[1] very coarse sand (1.0 to 0.5 mm), [2] coarse sand (0.5 to 0.25 mm), [3] medium sand (0.25 to 0.10 mm), [4] fine sand (0.1 mm to 0.05 mm), [5] very fine sand (0.05 to 0.002 mm); [6] HCP 1.0 (1.03 m), [7] PRP 1.1 (0.54 m), [8] HCP 2.0 (1.55 m), [9] PRP 2.1 (3.18 m), [10] Veris Shallow (0.3 m), [11] Veris Deep (0.9 m), [12] Topographic wetness index.

*2.4. Proximal Soil Sensing*

Soil EC$_a$ data was measured at six depths using two sensors, the DUALEM-21S (Dualem Inc., Milton, ON, Canada) and the Veris 3100 (Veris Technologies, Inc., Salina, KS, USA); this data was used to determine if one sensor or a particular sensing depth was more predictive of spatial variability or crop growth potential. Elevation data was simultaneously acquired with DUALEM measurements on 28–29 September 2016, using a real-time-kinematic GNSS system.

The depth of investigation of EC$_a$ measurements depended on the configuration of the transmitter and receiver coils. The DUALEM-21S has one transmitter coil and four receiving coils to capture four depths. Receiving coil spacing and arrangement affects the depth investigation. Receiver coils closer to the transmitter have a shallower depth of investigation. Additionally, coils arranged in the horizontal co-planar (HCP) receive lower depths than the perpendicular co-planar (PRP). The DUALEM-21S configuration

has two PRP coils spaced 1.1 m and 2.1 m from the transmitter (PRP1.1 and PRP 2.1, respectively), and two HCP coils spaced 1 m and 2 m from the transmitter (HCP1.0 and HCP2.0, respectively) [29,30]. A schematic overview of the of the DUALEM-21S sensor is presented in Figure 1. The DUALEM-21S was run for twenty minutes before being calibrated to reduce the possibility of drift in sensor data. It was pulled on a sled by a John Deere Gator at a relatively constant speed to maximize contact with the ground. At the end of sampling, the sensor was passed over previous transects so that data could be reviewed for evidence of drift.

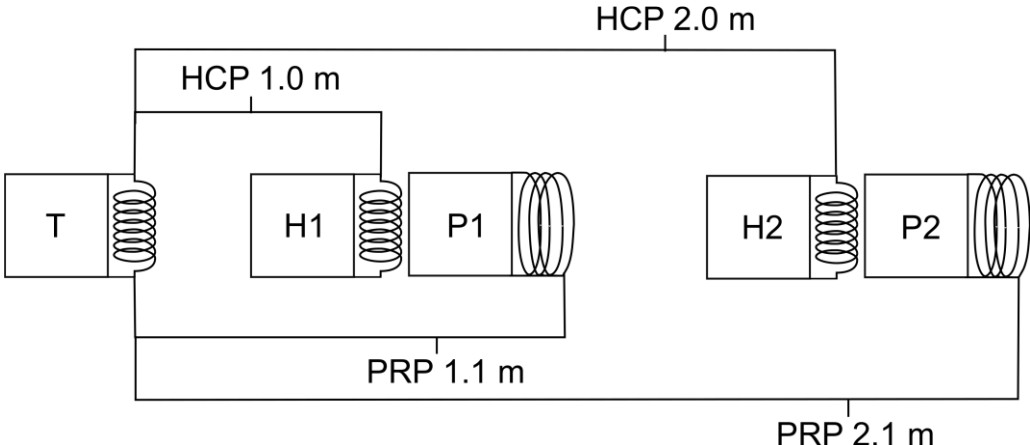

**Figure 1.** Schematic overview of the DUALEM-21S sensor with transmitting coil (T) and four receiving coils, two (H1 and H2) in horizontal coplanar (HCP) and two (P1 and P2) in a perpendicular (PERP) loop orientation [29,30].

The depth of investigation of the HCP 1.0 and HCP 2.0 were estimated to be about 1.55 m and 3.18 m, respectively; the depth of investigation of the depths PRP 1.1 and PRP 2.1 was estimated to be 0.54 m and 1.03 m, respectively. Depth of investigation was estimated where a 75% response is received by the sensor [31]. DUALEM/elevation transects were spaced approximately 10 m apart and sampled at a frequency of 1 Hz.

Veris $EC_a$ measurements were acquired on 21 October 2016. Effective sensing depths of the Veris Shallow and Veris Deep layers were 0.30 m and 0.90 m, respectively. The Veris 3100 is a galvanic contact resistivity sensor and derived conductivity from its inverse relationship with electrical resistivity. It was configured with six rolling coulter electrodes. Electrical current flows through the second and fifth coulters. The voltage drop is measured between the third and fourth coulters and first and sixth coulters (Sudduth et al., 2003). The electrodes are equally spaced in a Wenner array so that resistance is measured at two depths.

The Veris sensor transects were spaced approximately 3 m apart and sampled at a density of 1 measurement per second. Slope and topographic wetness index (TWI), ref. [32] were calculated from elevation data using SAGA GIS (v.2.1.2, System for Automated Geoscientific Analyses, Hamburg, Germany). The sensor data was prepared by first removing outliers outside two standard deviations followed by applying a moving average filter to reduce noise in the sensor data [33]. TWI is a commonly used hydrological index which describes the tendency of a given cell to accumulate water based on its catchment area and the slope angle [34]. It is defined as,

$$TWI = ln\left(\frac{\text{SCA}}{\tan \varphi}\right), \tag{1}$$

where SCA is the Specific Catchment Area and $\varphi$ is the slope angle. Higher TWI values are associated with greater water accumulation.

### 2.5. Satellite Imagery

Satellite imagery of the fields, on 11 August 2016, the same week that yield sampling was carried out, were obtained from Airbus's SPOT-6 satellite archive (Airbus Defense and Space, Ottobrunn, Germany). The SPOT-6 image was delivered georeferenced and corrected for off-nadir acquisition and terrain effects using the standard Reference 3-D model for ground corrections (Astrium Services, 2013). The image included red, green, blue, and near-infrared (NIR) bands at 5 m$^2$ resolution, as well as a panchromatic band at 1.5 m$^2$ resolution. It was pansharpened to 1.5 m$^2$ with the Gram-Schmidt method in ENVI image analysis software (Exelis Inc., Boulder, CO, USA), then radiometrically and atmospherically corrected. Pan-sharpening is an image-fusion technique that merges visible multispectral bands and the panchromatic band to produce color images of higher spatial resolution [35]. Pan-sharpening may affect the accuracy of color information in multi-spectral images, but the Graham-Schmidt method has been shown to improve spatial resolution with less effect on color reproduction [36,37]. For the purposes of this study, pan sharpening for a higher spatial resolution was prioritized over greater color accuracy, given the size of the wild blueberry bush clusters and bare spots.

Both study sites were subset from the pre-processed image and analyzed separately. Several broadband VIs known to estimate crop vigor and tree canopy in other studies were calculated from the multispectral bands (Table A1). In addition to broadband vegetation indices, principal component analysis (PCA) has been used in remote sensing to reduce dimensionality while preserving total variance of a dataset [38]. A smaller set of Principal components (PC) uncorrelated to one another are derived from the dataset, the first principal component (PC) representing the greatest proportion of variance, and the subsequent components accounting for the remaining variance of the original dataset [39]. PCA is frequently used in anomaly detection [40]. The second principal component (PC2) has been characterized as the "change component" which identifies seasonal changes in datasets [41,42]. Furthermore, the second principal component (PC2) has been found to distinguish different types of vegetation [43,44]. Because weeds and other vegetation may grow between wild blueberry bushes, the resulting PCs were compared with the VIs to examine if they could distinguish blueberry bush and bare spots.

The vegetation indices were calculated using the band math tool in ENVI software. A forward PCA Rotation was carried out in ENVI on a subset of the satellite image taken for each study site. A covariance matrix was calculated from the subset image. The number of output principal components was set to 4. The four principal components and an eigenvalue chart were generated. All the VI outputs and the principal components were exported as raster grids to ArcGIS. The values of each VI and PC raster were extracted at the geographic points where fruit yield and soil were sampled.

Pearson's correlation coefficient was calculated for each VI and the sampled yield in kg ha$^{-1}$ to evaluate the relationship of the VIs to fruit yield. Sampled yield was also classified as bare or not bare, where all yield values which were 0 kg ha$^{-1}$ were given a value of 0 and all yield values > 0 kg ha$^{-1}$ were assigned a value of 1. Again, Pearson's correlation was calculated for each of the VIs and PC2s with the classified yield.

### 2.6. Statistical Analysis

All sample and proximal sensor data were normalized using the lambda function of the 'box-cox' package in R (R Foundation for Statistical Computing, Vienna, Austria). The function automatically selected a box-cox transformation parameter ($\lambda$) which minimized the coefficient of variation of the data series [45–47]. Classical statistics including mean, standard deviation (*SD*), and coefficient of variation (*CV*) were calculated. Previous research has used *CV* as a first approximation of field heterogeneity [48]. The percentage of the coefficient of variation (*CV*) was used to evaluate the intensity of the variability of the datasets using the approach of Nolin and Caillier (1992), where *CV* is classified as weak (<15%), moderate (15–35%), strong (35–50%), very strong (50–100%), or extremely strong ($\geq$100%) [49].

Elevation and EC$_a$ data were interpolated to continuous surfaces using the Ordinary kriging method in ArcGIS (ESRI, Redlands, CA, USA). The values of topographic attributes (elevation, slope, TWI) and EC$_a$ at six depths (i.e., Veris Shallow, Veris Deep; DUALEM PRP 1.1, PRP 2.1, HCP 1.0, and HCP 2.0) were extracted from their respective kriging-interpolated surfaces at the same locations as the fruit yield and chemical/granulometric soil sampling points to compare and quantify the relationship between sensor data, soil properties, and fruit yield. The correlation coefficient (r) between soil attributes, fruit yield and sensor data was determined with Pearson's Correlation.

Spatial statistics were calculated using the 'gstat' package in R for all soil properties, fruit yield, and sensor data to assess spatial variability with R statistical software [50,51]. A theoretical variogram model (Pure nugget, spherical or exponential) was fitted to the experimental variogram of the box-cox transformed data. The corresponding nugget, range, partial sill, and total sill were calculated from the best-fitting variogram model. The degree of spatial dependence was classified using the Cambardella et al. (1994) approach whereby the nugget-to-sill ratio represented strong (<25%), moderate (25–75%), weak (>75%), or random (100%) spatial dependence [52]. Interpolated maps were produced for each of the properties using the Ordinary Kriging (OK) method to assess patterns of spatial variability. Accuracy of the maps were cross-validated with the leave-one-out method, and standardized root mean square error (RMSE$_r$) and coefficient of determination (R$^2$) were calculated for each map for comparison.

### 2.7. Selection and Comparison of Theoretical Combinations

The sensor layers (EC$_a$ and elevation) and their derivatives (slope and TWI) were compared by their correlation coefficients (Pearson's r) with soil properties to determine which datasets would be selected for the theoretical combinations of site-specific management scenarios. Among the six EC$_a$ depths, both DUALEM PRP 1.1 and Veris Shallow depths demonstrated significant correlation with the greatest number of sampled soil attributes in both fields (Tables 2 and 3). Similarly, elevation was shown to relate to more soil properties than to the topographic wetness index or slope, neither of which showed significant correlation with most soil properties in either field. Geostatistical analysis revealed moderate to strong spatial dependence of elevation, Veris Shallow, and DUALEM PRP 1.1 in both fields, indicating it would be feasible to characterize within-field variation with the sensor layers (Table 4). Furthermore, the kriging-interpolated maps displayed distinct within-field spatial patterns between elevation and EC$_a$, suggesting a combination of the two provided a better reflection of field conditions (Figures A4 and A5). Veris Shallow and DUALEM PRP 1.1 displayed similar spatial patterns on the kriging-interpolated maps, suggesting either sensor could be used to characterize within-field spatial variability (Figures A2 and A3). Ultimately, the shallowest sensing depth, Veris Shallow (0.3 m) was chosen with elevation for the theoretical combinations because it corresponded more closely with the rooting depth of wild blueberry (0–0.3 m).

Given that only two sensor layers were selected for theoretical combinations, a quadrant plot was used to identify four theoretical scenarios. Quadrant analysis is a simple method for classifying or sub-setting data by dividing an x-y scatterplot into four sections (quadrants). Quadrant plots have been used across many research disciplines to classify or subset datasets [53–55].

A scatterplot which projected elevation on the x-axis and Veris Shallow EC$_a$ on the y-axis was divided into four quadrants based on the median values of the dataset of EC$_a$ while topography was projected with the sample points (Figures 2 and 3). The upper-left quadrant represented sample points situated in low elevations with high EC$_a$ values. The lower-right quadrant represented points situated in high elevations with low EC$_a$ values.

**Table 2.** Pearson's correlation coefficient values of soil nutrients, fruit yield and sensor data in Field$_{Und}$.

| | Fruit Yield | Elevation | Slope | TWI [12] | HCP 1.0 [6] | PRP 1.1 [7] | HCP 2.0 [8] | PRP 2.1 [9] | Shallow [10] | Deep [11] |
|---|---|---|---|---|---|---|---|---|---|---|
| Fruit Yield | | −0.25 ** | n.s. | n.s. | 0.21 * | n.s. | 0.22 * | n.s. | n.s. | n.s. |
| **Soil attributes 0–0.05 m depth** | | | | | | | | | | |
| Total C | 0.44 *** | n.s. | n.s. | n.s. | n.s. | n.s. | n.s. | n.s. | n.s. | n.s. |
| Total N | 0.47 *** | n.s. | n.s. | n.s. | n.s. | n.s. | n.s. | n.s. | n.s. | n.s. |
| Soil pH | −0.21 * | −0.33 *** | 0.22 ** | n.s. | n.s. | 0.43 *** | 0.32 *** | 0.43 *** | 0.35 *** | 0.46 *** |
| P | −0.21 * | −0.29 *** | 0.28 ** | n.s. | n.s. | 0.38 *** | 0.24 ** | 0.38 *** | 0.35 *** | 0.37 *** |
| K | 0.46 *** | n.s. | n.s. | n.s. | n.s. | n.s. | n.s. | n.s. | n.s. | n.s. |
| Ca | n.s. | n.s. | n.s. | n.s. | n.s. | n.s. | n.s. | n.s. | n.s. | n.s. |
| Mg | 0.27 ** | n.s. | n.s. | n.s. | n.s. | 0.20 * | n.s. | 0.21 * | 0.20 * | n.s. |
| Al | −0.21 * | n.s. | n.s. | n.s. | n.s. | n.s. | n.s. | n.s. | n.s. | n.s. |
| Fe | n.s. | −0.20 * | n.s. | n.s. | n.s. | 0.34 *** | 0.23 ** | 0.34 *** | 0.30 *** | 0.23 ** |
| P/Al ratio | n.s. | −0.31 *** | 0.27 ** | n.s. | n.s. | 0.39 *** | 0.27 ** | 0.42 *** | 0.34 *** | 0.38 *** |
| **Soil attributes 0.05–0.15 m depth** | | | | | | | | | | |
| Total clay | n.s. | 0.55 *** | n.s. | n.s. | −0.18 * | −0.47 *** | −0.45 *** | −0.53 *** | −0.28 *** | −0.39 *** |
| Total silt | 0.19 * | −0.69 *** | n.s. | −0.18 * | 0.25 ** | 0.54 *** | 0.61 *** | 0.66 *** | 0.38 *** | 0.30 *** |
| Total sand | n.s. | 0.62 *** | n.s. | 0.17 * | −0.24 ** | −0.53 *** | −0.58 *** | −0.62 *** | −0.36 *** | −0.31 *** |
| Very coarse sand [1] | n.s. | 0.33 *** | n.s. | n.s. | n.s. | −0.35 *** | −0.37 *** | −0.38 *** | −0.22 * | −0.23 ** |
| Coarse sand [2] | n.s. | 0.63 *** | n.s. | n.s. | −0.22 * | −0.64 *** | −0.62 *** | −0.69 *** | −0.44 *** | −0.45 *** |
| Medium sand [3] | n.s. | 0.73 *** | n.s. | n.s. | −0.18 * | −0.66 *** | −0.62 *** | −0.71 *** | −0.45 *** | −0.47 *** |
| Fine sand [4] | n.s. | −0.22 * | n.s. | n.s. | n.s. | 0.28 *** | 0.19 * | 0.25 ** | 0.26 ** | 0.34 *** |
| Very fine sand [5] | n.s. | −0.74 *** | n.s. | n.s. | 0.21 * | 0.68 *** | 0.65 *** | 0.74 *** | 0.48 *** | 0.49 *** |

[1] very coarse sand (1.0 to 0.5 mm), [2] coarse sand (0.5 to 0.25 mm), [3] medium sand (0.25 to 0.10 mm), [4] fine sand (0.1 mm to 0.05 mm), [5] very fine sand (0.05 to 0.002 mm); [6] HCP 1.0 (1.03 m), [7] PRP 1.1 (0.54 m), [8] HCP 2.0 (1.55 m), [9] PRP 2.1 (3.18 m), [10] Veris Shallow (0.3 m), [11] Veris Deep (0.9 m), [12] Topographic Wetness Index; Correlation significance denoted by *, ** and ***, are equivalent to $p = 0.05$, $p = 0.01$ and $p = 0.001$, respectively; n.s. where non-significant at $p = 0.05$.

**Table 3.** Pearson's correlation coefficient values of soil nutrients, fruit yield and sensor data in Field$_{Flat}$.

| | Fruit Yield | Elevation | Slope | TWI [12] | HCP 1.0 [6] | PRP 1.1 [7] | HCP 2.0 [8] | PRP 2.1 [9] | Shallow [10] | Deep [11] |
|---|---|---|---|---|---|---|---|---|---|---|
| Fruit Yield | | n.s. | n.s. | n.s. | 0.20 * | n.s. | n.s. | 0.31 *** | n.s. | n.s. |
| **Soil attributes 0–0.05 m depth** | | | | | | | | | | |
| Total C | 0.38 *** | n.s. | n.s. | n.s. | n.s. | 0.33 *** | n.s. | 0.43 *** | 0.40 *** | n.s. |
| Total N | 0.39 *** | −0.19 * | n.s. | n.s. | n.s. | 0.38 *** | n.s. | 0.46 *** | 0.44 *** | n.s. |
| Soil pH | −0.21 * | −0.29 ** | n.s. | n.s. | n.s. | n.s. | n.s. | n.s. | n.s. | n.s. |
| P | n.s. | −0.31 *** | n.s. | n.s. | n.s. | n.s. | 0.28 ** | n.s. | n.s. | n.s. |
| K | 0.35 *** | n.s. | n.s. | n.s. | n.s. | n.s. | n.s. | 0.32 *** | n.s. | −0.20 * |
| Ca | 0.18 * | n.s. | n.s. | n.s. | 0.31 *** | n.s. | 0.41 *** | n.s. | n.s. | n.s. |
| Mg | 0.31 *** | −0.25 ** | n.s. | n.s. | n.s. | 0.32 *** | n.s. | 0.38 *** | 0.39 *** | n.s. |
| Al | n.s. | −0.29 ** | n.s. | n.s. | 0.25 ** | n.s. | 0.43 *** | n.s. | n.s. | 0.25 ** |
| Fe | n.s. | −0.22 * | −0.18 * | n.s. | n.s. | 0.29 ** | n.s. | 0.20 * | 0.42 *** | n.s. |
| P/Al ratio | n.s. | −0.28 ** | n.s. | n.s. | n.s. | n.s. | n.s. | n.s. | n.s. | n.s. |
| **Soil attributes 0.05–0.15 m depth** | | | | | | | | | | |
| Total clay | n.s. | n.s. | −0.22 * | n.s. | n.s. | −0.26 ** | n.s. | n.s. | n.s. | n.s. |
| Total silt | 0.35 *** | −0.20 * | n.s. | n.s. | n.s. | 0.31 *** | n.s. | 0.41 *** | 0.34 *** | n.s. |
| Total sand | −0.26 ** | n.s. | n.s. | n.s. | n.s. | −0.25 ** | n.s. | −0.28 ** | −0.32 *** | n.s. |
| Very coarse sand [1] | n.s. | n.s. | n.s. | n.s. | n.s. | n.s. | n.s. | n.s. | n.s. | −0.29 ** |
| Coarse sand [2] | n.s. | 0.38 *** | n.s. | n.s. | n.s. | −0.35 *** | −0.19 * | n.s. | −0.33 *** | −0.31 *** |
| Medium sand [3] | n.s. | 0.53 *** | 0.27 ** | n.s. | n.s. | −0.46 *** | −0.28 ** | −0.23 * | −0.51 *** | n.s. |
| Fine sand [4] | n.s. | −0.52 *** | −0.22 * | n.s. | n.s. | 0.36 *** | 0.23 * | n.s. | 0.39 *** | 0.22 * |
| Very fine sand [5] | n.s. | −0.52 *** | n.s. | 0.19 * | n.s. | 0.51 *** | 0.34 *** | 0.24 * | 0.49 *** | 0.20 * |

[1] very coarse sand (1.0 to 0.5 mm), [2] coarse sand (0.5 to 0.25 mm), [3] medium sand (0.25 to 0.10 mm), [4] fine sand (0.1 mm to 0.05 mm), [5] very fine sand (0.05 to 0.002 mm); [6] HCP 1.0 (1.03 m), [7] PRP 1.1 (0.54 m), [8] HCP 2.0 (1.55 m), [9] PRP 2.1 (3.18 m), [10] Veris Shallow (0.3 m), [11] Veris Deep (0.9 m), [12] Topographic Wetness Index; Correlation significance denoted by *, ** and ***, are equivalent to $p = 0.05$, $p = 0.01$ and $p = 0.001$, respectively; n.s. where non-significant at $p = 0.05$.

**Table 4.** Summary of spatial statistics including the range of influence, nugget ratio and degree of spatial dependence as classified by Cambardella et al. (1994) in $Field_{Und}$ and $Field_{Flat}$ experimental fields in Normandin, Quebec.

| | $Field_{Und}$ | | | | $Field_{Flat}$ | | | |
|---|---|---|---|---|---|---|---|---|
| | Range (m) | Nugget Ratio [x] (%) | Spatial Class [y] | $R^2$ | Range (m) | Nugget Ratio [x] (%) | Spatial Class [y] | $R^2$ |
| Soil attributes 0–0.05 m depth | | | | | | | | |
| Total N | - | 1.00 | R | - | 87 | 0.59 | M | 0.12 |
| Total C | - | 1.00 | R | - | 62 | 0.66 | M | 0.08 |
| Soil $pH_{water}$ | 82 | 0.40 | M | 0.31 | 104 | 0.58 | M | 0.24 |
| P | 17 | 0.27 | M | 0.10 | 557 | 0.00 | S | 0.35 |
| K | - | 1.00 | R | - | - | 1.00 | R | - |
| Ca | - | 1.00 | R | - | 279 | 0.66 | M | 0.11 |
| Mg | 8 | 0.00 | S | 0.01 | - | 1.00 | R | - |
| Al | 58 | 0.64 | M | 0.13 | 40 | 0.00 | S | 0.40 |
| Fe | - | 1.00 | R | - | 86 | 0.72 | M | 0.14 |
| P/Al ratio | 80 | 0.71 | M | 0.40 | - | 1.00 | R | - |
| Soil attributes 0.05–0.15 m depth | | | | | | | | |
| Total clay | - | 1.00 | R | - | 111 | 0.27 | M | 0.38 |
| Total silt | 5444 | 0.08 | S | 0.49 | - | 1.00 | R | - |
| Total sand | 479 | 0.40 | M | 0.46 | - | 1.00 | R | - |
| Very coarse sand [1] | 472 | 0.63 | M | 0.15 | 31 | 0.00 | S | 0.29 |
| Coarse sand [2] | 12242 | 0.01 | S | 0.42 | 216 | 0.15 | S | 0.62 |
| Medium sand [3] | 5352 | 0.04 | S | 0.65 | 107 | 0.21 | S | 0.60 |
| Fine sand [4] | 341 | 0.77 | W | 0.08 | 288 | 0.14 | S | 0.64 |
| Very fine sand [5] | 5564 | 0.02 | S | 0.72 | 274 | 0.06 | S | 0.74 |
| Fruit yield and sensor data | | | | | | | | |
| Fruit yield | - | 1 | R | - | - | 1.00 | R | - |
| Elevation | 87 | 0.01 | S | 1.00 | 75 | 0.00 | S | 0.99 |
| Veris Shallow [10] | 129 | 0.63 | M | 0.53 | 60 | 0.03 | S | 0.49 |
| Veris Deep [11] | 132 | 0.72 | M | 0.20 | 60 | 0.03 | S | 0.06 |
| PRP1.1 [7] | 129 | 0.32 | M | 0.45 | 96 | 0.32 | M | 0.15 |
| PRP2.1 [9] | 127 | 0.23 | S | 0.67 | 94 | 0.78 | W | 0.10 |
| HCP1.0 [6] | 121 | 0.13 | S | 0.85 | 126 | 0.65 | M | 0.76 |
| HCP2.0 [8] | 87 | 0.00 | S | 0.90 | 60 | 0.03 | S | 0.66 |

[x]: Nugget ratio = (nugget variance/total variance) × 100; [y]: S = strong spatial dependence (<25%); M = moderate spatial dependence (25–75%); W = weak spatial dependence (>75%); and R = random spatial dependence (100%) [52]. [1] very coarse sand (1.0 to 0.5 mm), [2] coarse sand (0.5 to 0.25 mm), [3] medium sand (0.25 to 0.10 mm), [4] fine sand (0.1 mm to 0.05 mm), [5] very fine sand (0.05 to 0.002 mm); [6] HCP 1.0 (1.03 m), [7] PRP 1.1 (0.54 m), [8] HCP 2.0 (1.55 m), [9] PRP 2.1 (3.18 m), [10] Veris Shallow (0.3 m), [11] Veris Deep (0.9 m).

A subset of the four theoretical combinations were created by selecting the 15 points from each quadrant which were situated at the outermost corner of the scatterplot. To ensure the subsets represented the average soil conditions in their respective theoretical combinations, the 10 points which showed the greatest similarity in sampled soil properties and yield were selected for each quadrant. Quadrants were delineated according to median values in the x and y datasets. $Field_{Und}$ displayed a bimodal distribution in $EC_a$ data, so two medians were calculated for $EC_a$ in the high elevation region and low elevation region, respectively (Figure 2).

To compare the four scenarios for significant differences, a Two-way Analysis of Variance (ANOVA) was calculated. Tukey–Kramer's post hoc test was performed to compare the four scenarios and determine the degree of separability of yield-limiting properties between the four theoretical combinations.

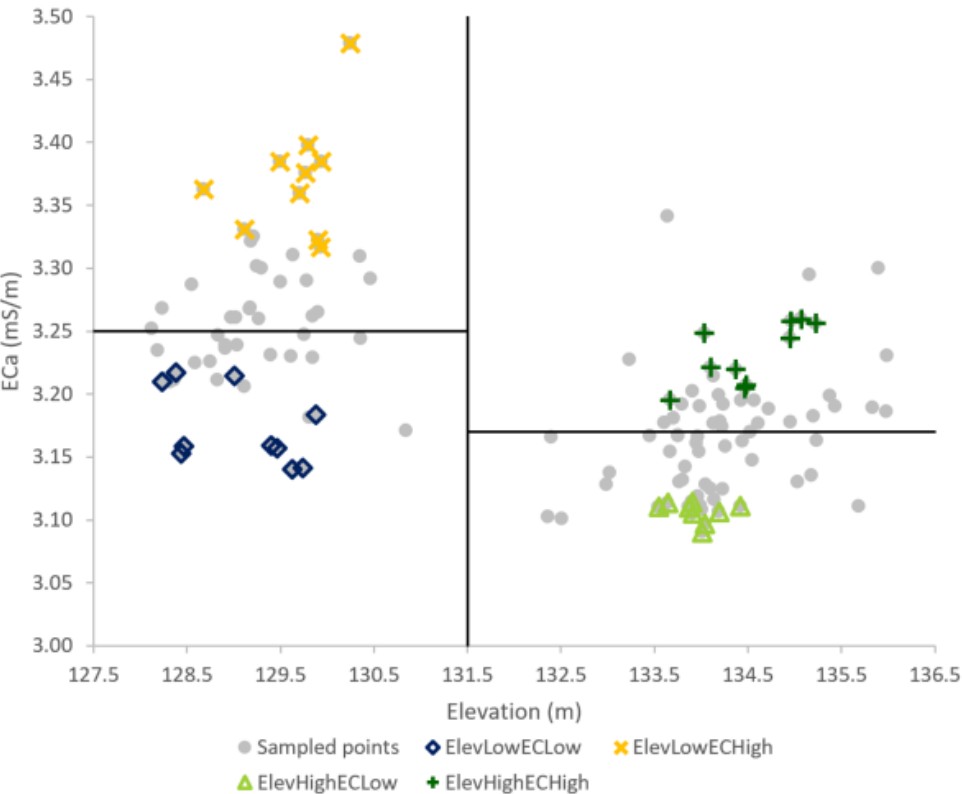

**Figure 2.** Scatter plot Veris Shallow EC$_a$ in Field$_{Und}$ used to select four distinct sensor combinations from the four quadrants of Field$_{Und}$.

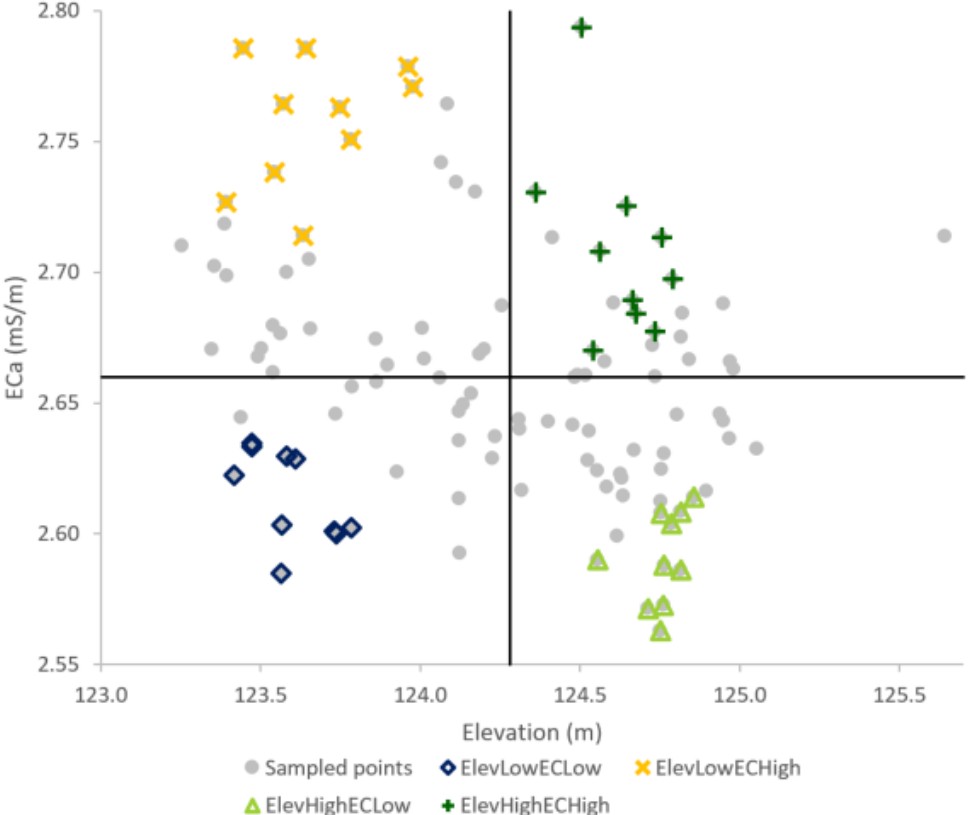

**Figure 3.** Scatter plot Veris Shallow EC$_a$ in Field$_{Flat}$ used to select four distinct sensor combinations from the four quadrants of Field$_{Flat}$.

To compare average soil conditions of bare spots to average field conditions in Field$_{Und}$, the standardized measurement (Z) was calculated for each soil attribute. Bare spots were defined as sample points where no blueberries could be harvested in a 1 m$^2$ area. The standardized measurement was calculated as,

$$Z = \frac{observed\ value - sample\ mean}{sample\ mean} \tag{2}$$

Tukey–Kramer's post hoc test was performed again to compare the average soil attributes of the bare spot scenario and the four sensor combination scenarios in Field$_{Und}$. The methodology and workflow are summarized in a flowchart in Figure 4.

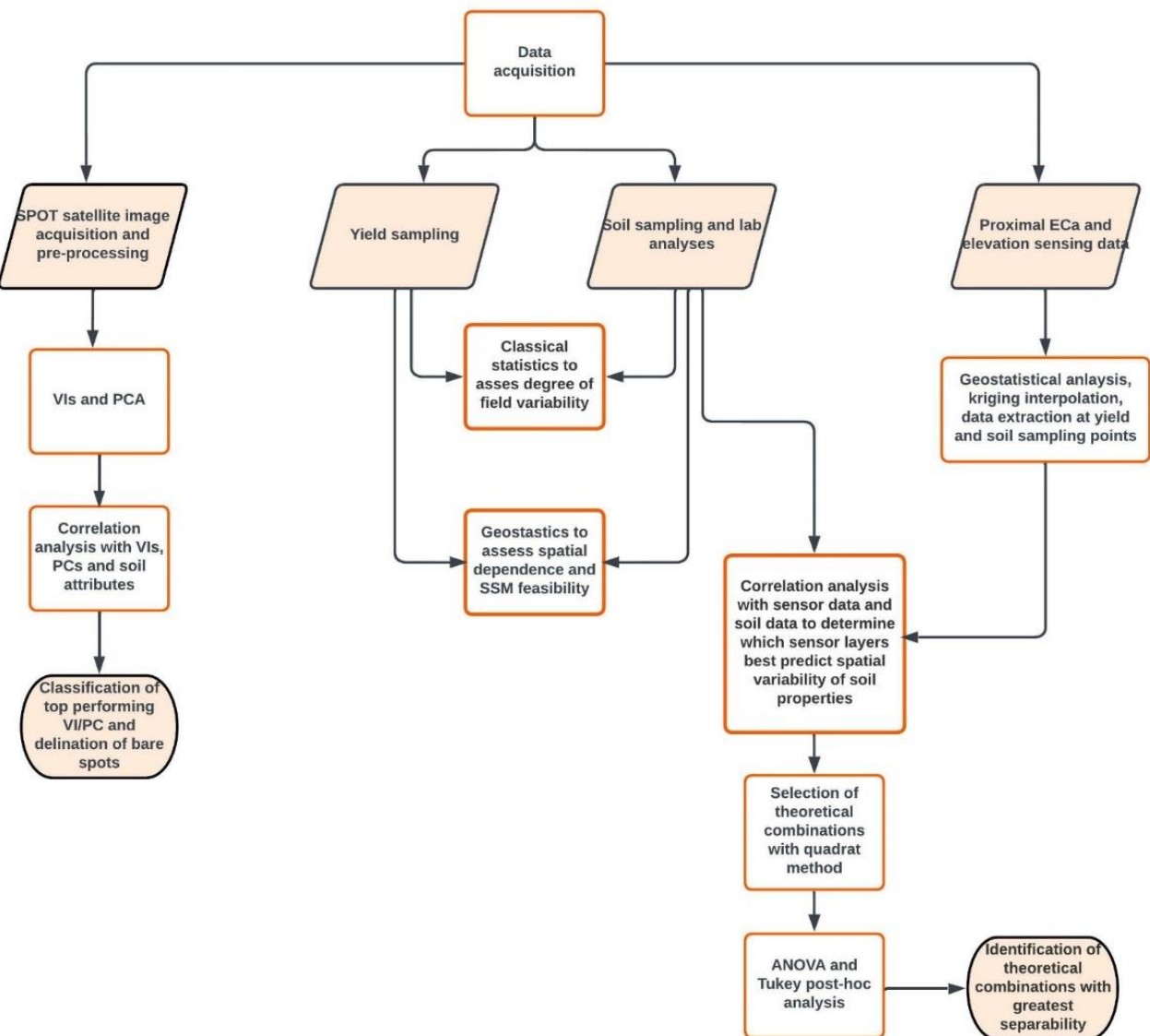

**Figure 4.** Flowchart representing the workflow and methodology.

## 3. Results and Discussion

### 3.1. Descriptive Statistics of Fruit Yield, Soil Properties and Sensor Data

For brevity, only the results from the chemical attributes at the 0–0.05 m depth and the physical attributes from the 0.05–0.15 m depth are discussed here. Descriptive statistics of soil attributes, fruit yield and sensor data are summarized in Table 1. Fruit yield showed very strong variability in both fields ($CV_{Und}$ = 54.4%, $CV_{Flat}$ = 56.5%). A study by Farooque et al. (2012) conducted in two wild blueberry fields in Nova Scotia reported

similar *CV* values of 49.52 % and 55.36% [4]. Most soil attributes also showed strong or very strong variability. In Field$_{Flat}$, P in the 0–0.05 m depth showed extremely strong variability with a *CV* value of 124%.

The high variability observed in yield and yield-determining attributes demonstrated field heterogeneity of the wild blueberry fields. High *CV* values of both fruit yield and soil attributes in Field$_{Und}$ and Field$_{Flat}$ may be explained by extrinsic sources, such as crop management or weather, or intrinsic sources such as natural variations in soil [56] and plant density [57]. The *CV* values of pH showed weak variability ($CV_{Und}$ = 10.6%, $CV_{Flat}$ = 7.8%). This may be due to the logarithmic scale of pH and has been observed in other studies [4,58]. Average soil pH was acidic (4.7), as is characteristic of Podzolic soils [21].

Silt content and sand partitioning were better indicators of variability in soil texture than total sand or total clay content. While *CV* was low for total sand content in both fields, total silt content showed high and very high variability ($CV_{Und}$ = 63%, $CV_{Flat}$ = 39%). Partitioning of sand furthermore revealed variability by sand grain size. In general, soil texture was more variable in Field$_{Und}$, likely due to the effects of varying topography on the accumulation of finer soils downslope and coarser soils upslope.

### 3.2. Relationship of Sensor Data to Fruit Yield and Soil Properties

A summary of Pearson's correlation coefficients of soil properties, fruit yield, sensor and topographic data are presented in Tables 2 and 3. In both fields, several soil attributes which research have shown to be yield-determining showed significant positive correlation with fruit yield, including total C ($r_{Und}$ = 0.44, $r_{Flat}$ = 0.38), total N ($r_{Und}$ = 0.47, $r_{Flat}$ = 0.39), K ($r_{Und}$ = 0.46, $r_{Flat}$ = 0.35), and Mg ($r_{Und}$ = 0.27, $r_{Flat}$ = 0.31). Total C represents the sum of organic and inorganic C in the soil and is directly related to the soil organic matter (SOM) content [59]. Higher total C at the soil surface could represent higher SOM, which in turn improves water holding capacity and nutrient availability [60,61]. Past research found total N to be the principle limiting nutrient for plant growth, fruit yield, and quality of wild blueberry in the Lac-Saint-Jean region [62]. Fertilization trials by Percival and Sanderson (2004) previously found main and interactive effects of soil-applied N, P, and K on stems per m$^2$ and specifically that soil-applied K influenced stem density and number of set fruit [63]. The relationship of fruit yield with nutrients at the 0–0.05 m depth may, in part, be due to the shallow rooting depth of wild blueberry (0.1–0.15 m), which could leave it sensitive to variations in weather and organic matter coverage [64,65].

In both fields, a negative correlation was observed between fruit yield and soil pH. Soil pH has been found to relate to foliar nutrient levels with the ideal pH range between 4 to 5 [66], and higher pH values are associated with weed growth which can limit blueberry stand growth. In Field$_{Und}$ the P/Al ratio, which can serve as an indicator of phosphorous accumulation, did not show significant correlation with yield, but did show significant correlation with EC$_a$ and elevation.

Of the topographic attributes, elevation showed significant correlation with more soil attributes in comparison to TWI or slope. In Field$_{Und}$, elevation was negatively correlated with fruit yield ($r_{Und}$ = −0.25) and strongly negatively correlated with total silt, fine sand, and very fine sand. In both fields, fruit yield was significantly positively correlated with total silt content ($r_{Und}$ = 0.19, $r_{Flat}$ = 0.35). Low lying regions of the field, where finer sands accumulate, would have improved water holding capacity and nutrient availability. Elevation also showed a significant negative relationship to pH, P, and Fe. In Field$_{Flat}$, elevation showed significant negative correlation with total N, pH, P, Mg, Al, and Fe. The findings demonstrate considerable correlation between elevation data and key soil attributes.

The correlation analysis indicated that both DUALEM PRP 1.1 (depth: 0.54 m) and Veris Shallow (depth: 0.3 m) similarly correlated with the majority of soil attributes, showing significant correlation with 12 of 17 attributes at the mineral depth. Higher correlation coefficients of soil attributes at these two depths can be explained in part by depth of soil sampling (0–0.15 m) and in part by the rooting zone of wild blueberries. Wild blueberry

rhizomes make up approximately 85% of wild blueberry [67] and are found in the first 0.15 m of soil, a majority of which are found in the first 0.10 m [64]. Nutrient storage and lateral water transport occur at this depth which may explain why the shallowest sensing depths showed a correlation to a greater number of chemical soil attributes. Again, findings demonstrate $EC_a$ to be an adequate predictor of key soil attributes.

*3.3. Spatial Structure of Soil Variability ($EC_a$ and Elevation)*

A summary of geostatistical parameters, including range, nugget-to-sill ratio, and spatial class as defined by Cambardella et al. (1994) is presented in Table 4 [52]. Additionally, the $R^2$ of the observed vs. predicted values are presented. Elevation showed strong spatial dependence in both fields. All $EC_a$ depths showed moderate to strong spatial dependence in both fields, except PRP 2.1 in $Field_{Flat}$ which showed weak spatial dependence. Fruit yield demonstrated random spatial dependence in both $Field_{Und}$ and $Field_{Flat}$.

Physical soil attributes demonstrated greater spatial dependence than chemical soil attributes. In $Field_{Und}$, certain chemical soil attributes showed moderate or strong spatial dependence, including soil pH, P, Mg, and Al. Soil texture attributes which showed moderate spatial dependence included total sand and very coarse sand. Total silt, coarse sand, medium sand, and very fine sand all showed strong spatial dependence in $Field_{Und}$. The same attributes all showed large range values in the variogram model, suggesting external trends in the field, or drift, such as elevation influence the soil texture [68,69]. Several soil attributes showed weak or random spatial dependence, including total N, total C, K, Ca, and Fe at the 0.0–0.05 m depth, and fine sand at the 0.05–0.15 m depth. Exogenous factors such as wind and sun exposure which introduce stochastic processes could have contributed to a lack of spatial structure for certain chemical attributes [70].

In $Field_{Flat}$, several physical attributes showed strong spatial dependence, total sand and total silt showed random spatial dependence, but the sand partitions (very coarse sand to very fine sand) all showed strong spatial dependence. At the 0–0.05 m depth in $Field_{Flat}$, total N, total C, pH, P, Ca, and Fe showed moderate spatial dependence and Al showed strong spatial dependence.

Contrary to $Field_{Und}$, in $Field_{Flat}$, more soil attributes showed spatial structure at the 0–0.05 m depth, suggesting that exogenous factors in the surface layer of the soil were not the only explanation for random spatial structure. A second explanation is that spatial dependence of fruit yield and many soil attributes occurred at a smaller range than the 33 m sampling interval [71]. Kerry and Oliver (2003) suggest to sample at one third the range of $EC_a$ data [4,72]. The practical ranges of $EC_a$ in $Field_{Und}$ varied between 87 m and 132 m, suggesting a sampling interval ~29–44 m. In $Field_{Flat}$, the practical ranges varied between 60 and 126 m, suggesting a sampling interval of ~15–20 m to capture spatial variability.

Attributes which showed no spatial dependence were not used for interpolation by kriging. Cross-validation of interpolated maps provided the root mean square error (RMSE) and coefficient of determination ($R^2$) values, which were used to evaluate the accuracy of each kriging-interpolated map. The RMSE was standardized ($RMSE_r$) by the total variation to compare among several variables. A $RMSE_r$ value >0.71 signifies that the kriging model accounted for less than 50% of variability at the validation points [73]. Strong spatial dependence did not always result in higher $R^2$ values or lower $RMSE_r$ values (e.g., very coarse sand content). In certain instances, a linear trend between observed and predicted values was observable in the cross-validation plots, but the spread of data resulted in high RMSE and low $R^2$ values. The $R^2$ was generally higher for kriging-interpolated maps of physical attributes than chemical attributes. Again, a greater sampling density may have better captured chemical processes and yielded more accurate maps.

Patterns of spatial variability were observed in the kriging-interpolated maps. Certain attributes followed the same pattern as elevation, while others showed patterns more like $EC_a$. Others presented unique patterns. The spatial dependence observed among several agronomic properties (soil texture, pH, P, total N, and total C) and their relationship to elevation and $EC_a$ reaffirmed the viability of site-specific management.

### 3.4. Characterization and Delineation of Bare Spots

The standardized values, average soil conditions in bare spots, and average field conditions are presented in Table 5. In Field$_{Und}$, sampled bare spots occurred in the high elevation region and were mostly distributed among areas of higher EC$_a$, suggesting that high EC$_a$ alone is not a predictor of fruit yield and reiterating the need to treat bare spots separately from other parts of the field. Inversely, in Field$_{Flat}$, sampled bare spots occurred in an area of low elevation, low EC$_a$. The average soil conditions in bare spots had a slightly lower EC$_a$ (Z$_{Und}$ = −0.14 Z$_{Flat}$ = −0.09), yet soil showed higher than average pH (Z$_{Und}$ = 0.33, Z$_{Flat}$ = 2.53), and lower than average total C (Z$_{Und}$ = −0.27, Z$_{Flat}$ = −1.19), K (Z$_{Und}$ = −0.59, Z$_{Flat}$ = −1.347), and Ca (Z$_{Und}$ = −0.25, Z$_{Flat}$ = −1.73). This suggested soil conditions in bare spots differed considerably from average field conditions, despite little change in EC$_a$. In Field$_{Und}$ bare spots occurred at above average elevations (Z$_{und}$ = 1.1) and slope (Z$_{und}$ = 0.68), while in Field$_{Flat}$ bare spots occurred in below average elevations (Z$_{flat}$ = −1.16) and above average slope (Z$_{flat}$ = 0.47).

**Table 5.** Statistical comparison of key soil properties in bare spots relative to field averages Field$_{Und}$ and Field$_{Flat}$ experimental fields in Normandin, Quebec. Standardized measurements (Z) of soil properties close to a value of 0 resemble the field average.

| | Field$_{Und}$ | | | Field$_{Flat}$ | | |
|---|---|---|---|---|---|---|
| | Z Score [13] | Bare Spot Average | Field Average | Z Score [13] | Bare Spot Average | Field Average |
| Soil attributes 0–0.05 m depth | | | | | | |
| Total Carbon (C) | −1.14 | 3.64 | 11.1 | −1.24 | 2.53 | 8.80 |
| Total Nitrogen (N) | −1.17 | 0.14 | 0.46 | −1.16 | 0.15 | 0.44 |
| Soil pH$_{water}$ | 0.91 | 5.16 | 4.73 | 2.29 | 5.34 | 4.50 |
| Phosphorous (P) | 0.54 | 92.0 | 63.4 | 1.20 | 96.1 | 39.0 |
| Potassium (K) | −1.04 | 33.9 | 107 | −1.14 | 28.9 | 93.0 |
| Calcium (Ca) | −0.34 | 335 | 361 | −1.46 | 242 | 387 |
| Magnesium (Mg) | −0.82 | 48.3 | 107 | −1.09 | 19.4 | 78.0 |
| Aluminum (Al) | 1.35 | 1277 | 889 | 0.52 | 1093 | 939 |
| Iron (Fe) | −0.40 | 1126 | 1502 | −0.80 | 193 | 465 |
| P/Al ratio | 0.30 | 0.08 | 0.069 | 1.23 | 0.09 | 0.039 |
| Soil attributes 0.05–0.15 m depth | | | | | | |
| Total Clay | 0.33 | 25.3 | 23.6 | −1.57 | 16.9 | 26.5 |
| Total Silt | −0.59 | 74.8 | 120 | −1.12 | 43.3 | 77.5 |
| Total Sand | 0.58 | 900 | 857 | 1.45 | 940 | 896 |
| Very coarse sand [1] | −0.38 | 6.85 | 12.0 | 0.22 | 28.8 | 25.4 |
| Coarse sand [2] | 0.01 | 101 | 99.9 | −0.64 | 113 | 170 |
| Medium sand [3] | 0.44 | 356 | 285 | −0.64 | 291 | 357 |
| Fine sand [4] | 0.41 | 362 | 312 | 1.32 | 447 | 280 |
| Very fine sand [5] | −0.57 | 73.6 | 148 | −0.06 | 60.2 | 63.3 |
| Fruit yield and sensor data | | | | | | |
| HCP 1.0 [6] | −0.64 | 3.90 | 4.31 | −1.38 | 3.91 | 4.26 |
| PRP 1.1 [7] | −0.46 | 1.28 | 1.33 | 0.168 | 1.03 | 1.02 |
| HCP 2.0 [8] | −0.88 | 3.59 | 3.84 | −0.73 | 2.84 | 2.95 |
| PRP 2.1 [9] | −0.94 | 1.58 | 1.65 | −1.02 | 1.25 | 1.31 |
| Veris Shallow [10] | −0.14 | 3.20 | 3.21 | −0.09 | 2.66 | 2.70 |
| Veris Deep [11] | 0.08 | 2.89 | 2.86 | −1.25 | 1.77 | 2.30 |
| Elevation | 1.09 | 135 | 132 | −1.16 | 123.6 | 124.3 |
| Fruit yield | −1.84 | 0 | 643 | −1.77 | 0 | 399 |
| Slope | 0.72 | 3.75 | 1.90 | 0.47 | 1.30 | 0.90 |
| TWI [12] | −0.371 | 5.22 | 6.40 | 0.14 | 5.50 | 5.00 |

[1] very coarse sand (1.0 to 0.5 mm), [2] coarse sand (0.5 to 0.25 mm), [3] medium sand (0.25 to 0.10 mm), [4] fine sand (0.1 mm to 0.05 mm), [5] very fine sand (0.05 to 0.002 mm); [6] HCP 1.0 (1.03 m), [7] PRP 1.1 (0.54 m), [8] HCP 2.0 (1.55 m), [9] PRP 2.1 (3.18 m), [10] Veris Shallow (0.3 m), [11] Veris Deep (0.9 m), [12] Topographic wetness index, [13] Z = observed value—sample mean/sample standard deviation.

Pearson's correlation values of VIs and fruit yield are presented in Table 6. The VIs did not show a strong relationship with fruit yield, but several showed promise for classifying

bare spots. When fruit yield was classified in binary terms (bare vs. vegetation), the correlation coefficients of several VIs improved. Correlation between the VIs and Field$_{Und}$ was likely higher because bare spots in the field were larger and more contiguous than in Field$_{Und}$ (Figure 5).

**Table 6.** Pearson's Correlation of Vegetation Indices (VI).

| | Field$_{Flat}$ | | Field$_{Und}$ | |
|---|---|---|---|---|
| | **Fruit Yield** | **Bare Spots** | **Fruit Yield** | **Bare Spots** |
| Normalized Difference Vegetation Index (NDVI) | 0.07 | 0.04 | 0.23 | 0.46 |
| Transformed Difference Vegetation Index (TDVI) | 0.18 | 0.10 | 0.29 | 0.55 |
| Optimized Soil Adjusted Vegetation Index (OSAVI) | 0.15 | 0.04 | 0.19 | 0.40 |
| Non-Linear Index (NLI) | 0.11 | 0.01 | 0.16 | 0.34 |
| Modified Simple Ratio (MSR) | 0.25 | 0.11 | 0.26 | 0.50 |
| Green Ratio Vegetation Index (GRVI) | 0.25 | 0.12 | 0.22 | 0.44 |
| Green Difference Vegetation Index (GDVI) | 0.04 | 0.04 | 0.07 | 0.25 |
| Enhanced Vegetation Index (EVI) | −0.06 | −0.29 | −0.09 | 0.04 |
| Modified Soil Adjusted Vegetation Index (MSAVI2) | 0.08 | −0.01 | 0.13 | 0.34 |
| First Principal Component (PC1) | −0.08 | −0.06 | −0.18 | −0.21 |
| Second Principal Component (PC2) | −0.40 | −0.32 | −0.39 | −0.64 |
| Third Principal Component (PC3) | 0.12 | 0.07 | 0.01 | −0.08 |
| Fourth Principal Component (PC4) | −0.00 | 0.25 | −0.08 | −0.10 |
| PC2 classified | 0.24 | 0.40 | 0.40 | 0.68 |

As previous research suggested, the second principal component showed significantly greater correlation values with yield and bare spots than the other principal components. PC2 showed moderately strong negative correlation with both yield and bare spots. In Field$_{Und}$, correlation with PC2 and bare spots showed a significantly stronger relationship than the other VIs and PCs, whereas correlation with bare spots and yield were similar in Field$_{Flat}$.

Because PC2 demonstrated the strongest relationship to crop vigor ($r_{Und}$ = 0.40, $r_{Flat}$ = −0.39), Jenks optimization method was used to classify the PC2 raster into two classes with natural breaks in the histogram [74]. The two natural breaks coincided with the PC2 bare vs. vegetation. When fruit yield was classified bare vs. vegetation, the correlation coefficient of PC2 with bare spots improved to $r_{Und}$ = 0.68, $r_{Flat}$ = 0.40. Xu and Su (2017) mentioned that limitations to VI yield mapping exist in horticulture due to heterogeneous canopies of soils, weeds, and cover crops [75]. While PC2 distinguished bare soil and vegetation, the spatial resolution of the multispectral image did not permit us to investigate its ability to distinguish types of vegetation.

*3.5. Separability of Key Soil Properties among Theoretical Combinations (Scenarios)*

A bare spot scenario derived from the classified VI was compared to the four scenarios in Field$_{Und}$ to ascertain whether the integration of bare spots delineated from satellite imagery improved the separability of key soil properties. The bare spot scenario was not compared in Field$_{Flat}$ due to lack of a sufficient number of sample points located within bare spots.

Tukey–Kramer's test showed that the greatest number of significantly different yield-determining properties were observed between scenario Elev$_{Low}$EC$_{High}$ and scenario Elev$_{High}$EC$_{Low}$ in both fields (Tables 7 and 8). In Field$_{Und}$, six key soil properties were distinguished (pH at both depths, Fe, coarse sand, medium sand, and very fine sand at the mineral depth). Other attributes such as pH and medium sand content showed separability among multiple scenarios, suggesting that site-specific management based on the combination of sensors may separate distinct soil attributes.

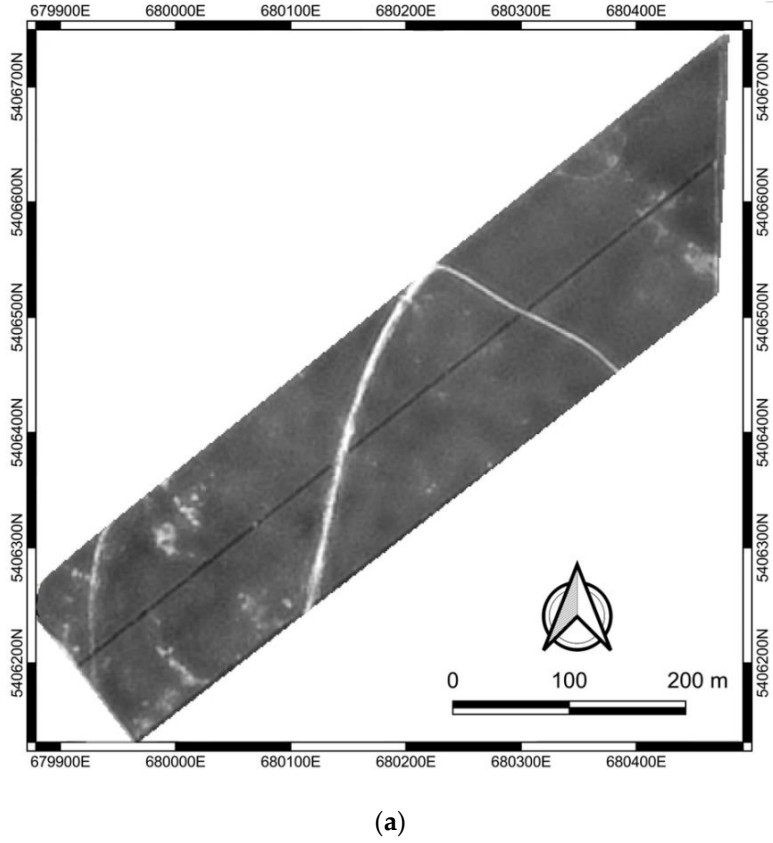

(**a**)

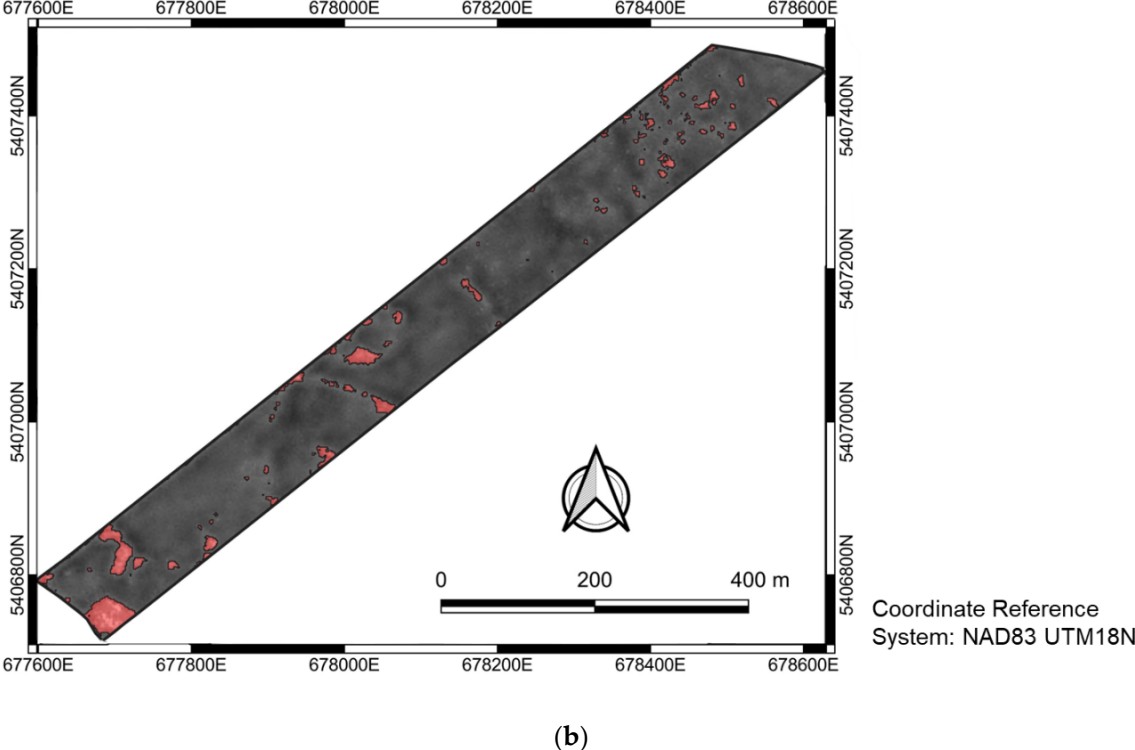

(**b**)

**Figure 5.** Map of the second principal component (PC2) in (**a**) Field$_{\text{Flat}}$ and map of PC2 with delineated bare spots in (**b**) Field$_{\text{Und}}$.

**Table 7.** Separability of key yield-determining properties among four distinct scenarios of combinations of Veris Shallow $EC_a$ and elevation in experimental field $Field_{Und}$. Values followed by different letters indicate significant differences according to Tukey–Kramer's test ($p > 0.05$). Properties highlighted in gray are those separated by the optimal combination of $Elev_{Low}EC_{High}$ and $Elev_{High}EC_{Low}$ or by the bare spot classification of the SPOT-6 satellite image.

| Property | Unit | $Elev_{Low} EC_{Low}$ | | $Elev_{Low} EC_{High}$ | | $Elev_{High} EC_{Low}$ | | $Elev_{High} EC_{High}$ | | Bare | |
|---|---|---|---|---|---|---|---|---|---|---|---|
| \multicolumn Soil attributes 0–0.05 m depth | | | | | | | | | | | |
| Total C | % | 9.69 | a | 13.1 | a | 9.24 | a | 13.9 | a | 6.93 | a |
| Total N | % | 0.42 | ab | 0.54 | ab | 0.39 | ab | 0.58 | a | 0.23 | b |
| pH | – | 4.72 | abc | 4.89 | ab | 4.38 | c | 4.61 | bc | 5.13 | a |
| P | mg kg$^{-1}$ | 61.4 | a | 81.5 | a | 51.7 | a | 39.9 | a | 101 | a |
| K | mg kg$^{-1}$ | 99.0 | ab | 120 | ab | 117 | ab | 163 | a | 60.0 | b |
| Ca | mg kg$^{-1}$ | 371 | a | 395 | a | 367 | a | 332 | a | 359 | a |
| Mg | mg kg$^{-1}$ | 89.9 | ab | 137 | ab | 79.1 | b | 171 | a | 66.8 | b |
| Al | mg kg$^{-1}$ | 880 | b | 906 | b | 831 | b | 763 | b | 1281 | a |
| Fe | mg kg$^{-1}$ | 1339 | a | 1995 | a | 1052 | a | 2003 | a | 1173 | a |
| P/Al ratio | – | 0.070 | a | 0.088 | a | 0.057 | a | 0.49 | a | 0.086 | a |
| \multicolumn Soil attributes 0.05–0.05 m depth | | | | | | | | | | | |
| Total C | % | 1.13 | a | 1.16 | a | 1.05 | a | 1.53 | a | 1.16 | a |
| Total N | % | 0.07 | a | 0.06 | a | 0.06 | a | 0.07 | a | 0.07 | a |
| pH | – | 5.13 | ab | 5.25 | a | 4.91 | b | 4.92 | b | 5.14 | ab |
| P | mg kg$^{-1}$ | 78.1 | a | 61.8 | a | 38.3 | a | 71.1 | a | 68.6 | a |
| K | mg kg$^{-1}$ | 30.2 | a | 34.9 | a | 40.1 | a | 45.5 | a | 28.7 | a |
| Ca | mg kg$^{-1}$ | 276.3 | ab | 330.6 | ab | 235 | b | 359 | a | 290 | ab |
| Mg | mg kg$^{-1}$ | 6.20 | a | 8.60 | a | 5.40 | a | 8.80 | a | 7.90 | a |
| Al | mg kg$^{-1}$ | 1742 | a | 1624 | a | 1749 | a | 1639 | a | 1657 | a |
| Fe | mg kg$^{-1}$ | 110 | ab | 216 | a | 60.2 | b | 152 | ab | 159 | ab |
| Total sand | g kg$^{-1}$ | 824 | a | 819 | a | 892 | a | 888 | a | 874 | a |
| Total silt | g kg$^{-1}$ | 154 | a | 159 | a | 81.2 | a | 84.7 | a | 101 | a |
| Total clay | g kg$^{-1}$ | 22.4 | a | 21.6 | a | 26.7 | a | 27.5 | a | 25.5 | a |
| Very coarse sand | g kg$^{-1}$ | 4.70 | a | 8.10 | a | 15.2 | a | 10.7 | a | 17.1 | a |
| Coarse sand | g kg$^{-1}$ | 75.4 | ab | 34.1 | b | 152 | a | 113 | ab | 116 | ab |
| Medium sand | g kg$^{-1}$ | 216 | bc | 143 | c | 395 | a | 391 | a | 293 | ab |
| Fine sand | g kg$^{-1}$ | 342 | a | 399 | a | 261 | a | 307 | a | 330 | a |
| Very fine sand | g kg$^{-1}$ | 185 | ab | 235 | a | 68.9 | b | 66.4 | b | 118 | ab |
| Fruit yield | g m$^{-2}$ | 717 | a | 632 | a | 671 | a | 543 | ab | 193 | b |
| TWI | – | 6.70 | a | 5.54 | a | 6.60 | a | 6.94 | a | 6.40 | a |
| Slope | deg | 1.50 | b | 1.22 | b | 0.31 | b | 4.79 | a | 2.99 | ab |

Furthermore, Tukey–Kramer's test in $Field_{Und}$ showed that the integration of classified bare spots significantly separated properties in the 0–0.05 m depth which could not be separated by the four theoretical scenarios. The bare spot scenario was most distinct from the scenario $Elev_{High}EC_{High}$. Results suggest a number of soil properties are separated between the $Elev_{High}EC_{High}$ scenario and bare spots (Total N, pH, K, Mg, and Al in the 0–0.05 m depth). These findings indicate that $Field_{Und}$, where bare spots were larger and more contiguous, would benefit from bare spot delineation with satellite imagery to reduce the use of chemicals and other site-specific management practices that could increase the profitability of wild blueberry production (Figure 6). Based on the classified image, 75.5 m$^2$ or 8.5% of $Field_{Und}$ was bare and 29.3 m$^2$ or 10.7% of $Field_{Flat}$ was bare. The percentages of bare spots was low compared to other studies which reported bare spots to be as high as 50% of blueberry fields [13,14].

**Table 8.** Separability of key yield-determining properties among four distinct scenarios of combinations of Veris Shallow $EC_a$ and elevation in experimental field $Field_{Flat}$. Values followed by different letters indicate significant differences according to Tukey–Kramer's test ($p > 0.05$). Properties highlighted in gray are those separated by the optimal combination of $Elev_{Low}EC_{High}$ and $Elev_{High}EC_{Low}$.

| Property | Unit | $Elev_{Low}$ $EC_{Low}$ | | $Elev_{Low}$ $EC_{High}$ | | $Elev_{High}$ $EC_{Low}$ | | $Elev_{High}$ $EC_{High}$ | |
|---|---|---|---|---|---|---|---|---|---|
| | | *Soil attributes 0–0.05 m depth* | | | | | | | |
| Total C | % | 9.3 | ab | 12.9 | a | 4.72 | b | 12.1 | a |
| Total N | % | 0.45 | ab | 0.70 | a | 0.24 | b | 0.59 | a |
| pH | – | 4.58 | a | 4.64 | a | 4.51 | a | 4.28 | a |
| P | $mg\,kg^{-1}$ | 40.6 | a | 70.4 | a | 26.4 | a | 24.9 | a |
| K | $mg\,kg^{-1}$ | 96.4 | ab | 99.8 | ab | 56.8 | b | 134 | a |
| Ca | $mg\,kg^{-1}$ | 456.9 | a | 376 | a | 378 | a | 386 | a |
| Mg | $mg\,kg^{-1}$ | 77.9 | ab | 113 | a | 36.3 | b | 124 | a |
| Al | $mg\,kg^{-1}$ | 990.6 | ab | 1232 | a | 899 | ab | 792 | b |
| Fe | $mg\,kg^{-1}$ | 454.7 | ab | 766 | a | 211 | b | 775 | a |
| P/Al ratio | – | 0.040 | a | 0.058 | a | 0.031 | a | 0.031 | a |
| | | *Soil attributes 0.05–0.15 m depth* | | | | | | | |
| Total C | % | 1.09 | ab | 1.37 | a | 0.87 | b | 1.19 | ab |
| Total N | % | 0.08 | ab | 0.10 | a | 0.07 | b | 0.08 | ab |
| pH | – | 5.08 | a | 5.05 | a | 4.96 | b | 4.90 | ab |
| P | $mg\,kg^{-1}$ | 27.2 | ab | 33.0 | a | 9.60 | b | 19.1 | ab |
| K | $mg\,kg^{-1}$ | 43.2 | ab | 46.5 | ab | 30.8 | b | 55.2 | a |
| Ca | $mg\,kg^{-1}$ | 233.3 | a | 217 | ab | 147 | b | 269 | a |
| Mg | $mg\,kg^{-1}$ | 7.9 | a | 8.1 | a | 5.00 | a | 7.2 | a |
| Al | $mg\,kg^{-1}$ | 1920.7 | a | 2057 | a | 1983 | a | 2101 | a |
| Fe | $mg\,kg^{-1}$ | 219.1 | ab | 324 | a | 87.3 | b | 277 | a |
| Total sand | $g\,kg^{-1}$ | 898.1 | ab | 868 | b | 910 | a | 890 | ab |
| Total silt | $g\,kg^{-1}$ | 76.9 | ab | 104 | a | 60.6 | b | 85.7 | ab |
| Total clay | $g\,kg^{-1}$ | 25.0 | a | 28.5 | a | 29.2 | a | 24.4 | a |
| Very coarse sand | $g\,kg^{-1}$ | 26.4 | a | 31.7 | a | 30.6 | a | 32.5 | a |
| Coarse sand | $g\,kg^{-1}$ | 143.4 | b | 126 | b | 278 | a | 195 | ab |
| Medium sand | $g\,kg^{-1}$ | 334.3 | ab | 224 | b | 443 | a | 336 | a |
| Fine sand | $g\,kg^{-1}$ | 330.0 | a | 373 | a | 135 | b | 247 | ab |
| Very fine sand | $g\,kg^{-1}$ | 63.9 | ab | 113 | a | 24.6 | b | 79.6 | ab |
| Fruit yield | $kg\,ha^{-1}$ | 3893 | a | 5487 | a | 3359 | a | 4755 | a |
| TWI | – | 4.41 | a | 6.04 | a | 4.97 | a | 5.90 | a |
| Slope | deg | 0.78 | a | 0.59 | a | 0.44 | a | 1.10 | a |

In $Field_{Flat}$, many (15) soil properties were distinguished between combinations $Elev_{Low}EC_{High}$ and $Elev_{High}EC_{Low}$. Notably, total C and total N were distinct at both depths. Certain properties (K at both depths, P at the mineral depth) were distinguished between combinations $Elev_{High}EC_{Low}$ vs. $Elev_{High}EC_{High}$, demonstrating some separability among these scenarios as well.

In both fields the scenario $Elev_{Low}EC_{High}$ showed significantly greater very fine sand content, total silt content, and yield-determining soil nutrients. The findings suggest this theoretical combination may be favorable to wild blueberry growth in these two studysites. Sand partitioning illustrated that finer textured soils were more favorable for wild blueberry yield. Further research to investigate crop response to fertilization between the identified scenarios is recommended.

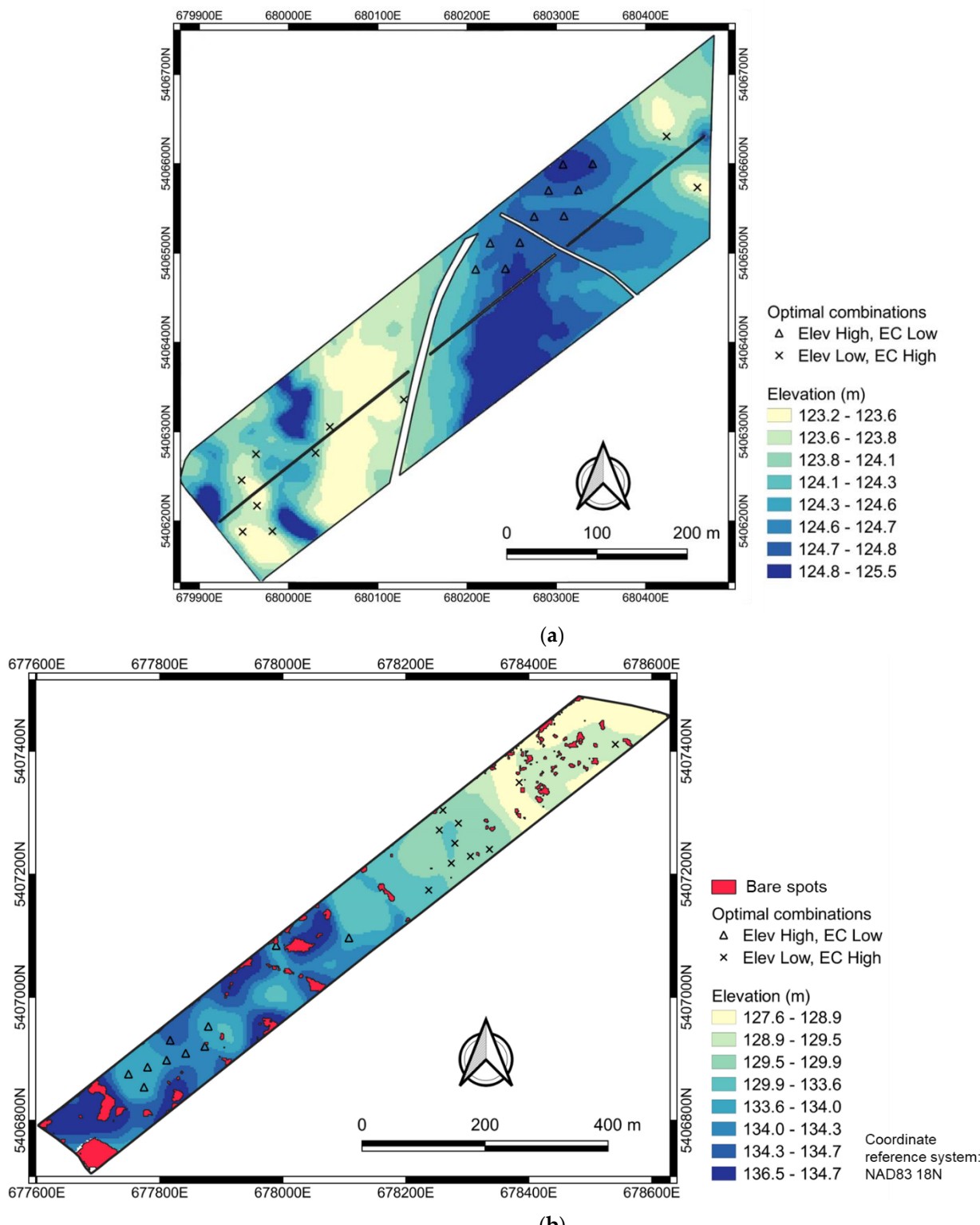

**Figure 6.** Theoretical combinations of field scenarios in (**a**) Field_Flat and (**b**) Field_Und. The classified bare spots serve as a third scenario in Field_Und where bare spots were larger and more contiguous.

All presented growing condition scenarios have been delineated to the size sufficient for the implementation of site-specific soil treatment plots to define optimum soil treatment conditions in each case. Since it is clear that at least two scenarios illustrate radically different production environments in a single field. It is reasonable to suspect different soil

management needs in these areas. Once established, such treatments could be applied to corresponding MZs while pursuing discrete prescription variable rate treatments. Alternatively, optimum management scenarios could be linked to high-density field elevation, shallow depth apparent soil electrical conductivity data as well as their combinations to derive continuous soil management prescription maps.

## 4. Conclusions

Results confirmed field heterogeneity of fruit yield in both study sites ($CV_{Und}$ = 54.4%, $CV_{Flat}$ = 56.5%) as well as a number of yield-determining soil attributes. However, yield did not show strong spatial dependence. A greater sampling density may better capture spatial dependence in yield and many chemical soil attributes (eg total C, total N, K).

Spatial variability of soils was best predicted with soil texture. Silt content and sand partitioning proved to be better indicators of variability in soil texture than total sand or total clay content. Total silt, coarse sand, medium sand, and very fine sand all showed strong spatial dependence in Field$_{Und}$, while sand partitioning (very coarse to very fine sand) showed the strongest spatial dependence in Field$_{Flat}$. Physical soil attributes were significantly correlated with both EC$_a$ and elevation, justifying the application of proximal sensors for a site-specific management approach.

Vegetation indices (VIs) obtained from satellite data showed promise as a biomass indicator, with the second principal component (PC2) showing the highest correlation with yield and bare spots in both study sites. In place of a clustering method, a quadrant analysis of the shallowest EC$_a$ depth vs. elevation provided four sensor combinations (scenarios) for theoretical field conditions. These theoretical combinations may be used to identify field conditions, rather than zones, requiring variable rate treatment. Prescription maps may be developed based on nutrient input needs of blueberry in the identified field conditions. ANOVA and Tukey–Kramer's post hoc test showed the greatest separability of soil properties in both fields were between the combinations of high EC$_a$/low elevation versus low EC$_a$/high elevation. In addition, the bare spots classified with satellite imagery in Field$_{Und}$ showed significantly distinct soil properties. Combining elevation, proximal and multispectral data predicted within-field variation of yield-determining soil properties and offered three theoretical scenarios in Field$_{Und}$ (high EC$_a$/low elevation; low EC$_a$/high elevation; bare spots) on which to base site-specific management and two theoretical scenarios (high EC$_a$/low elevation; low EC$_a$/high elevation) in Field$_{Flat}$. Future studies should investigate crop response to fertilization between the identified scenarios. Future research directions may also be highlighted.

**Author Contributions:** Conceptualization, A.N.C., J.L. (Jean Lafond), A.J. and V.A.; methodology, A.J., V.A., A.B. and A.N.C.; statistical analysis, A.J., V.A. and I.P.; writing—original draft preparation, A.J.; writing—review and editing, A.N.C., V.A., A.B., J.L. (Jean Lafond), J.L. (Julie Lajeunesse) and M.D.; visualization, A.J.; supervision, A.N.C. and V.A; project administration, A.N.C. and J.L. (Jean Lafond); funding acquisition, A.N.C. and J.L. (Jean Lafond). All authors have read and agreed to the published version of the manuscript.

**Funding:** This research was funded by Agriculture and Agri-food Canada (AAFC), grant number J-001401.

**Institutional Review Board Statement:** Not applicable.

**Informed Consent Statement:** Not applicable.

**Data Availability Statement:** Not applicable.

**Acknowledgments:** The authors recognize Denis Bourgeault, Sarah-Maude Parent, and Claude Lévesque of AAFC for their field work and chemical analysis laboratory work.

**Conflicts of Interest:** The authors declare no conflict of interests.

**Appendix A**

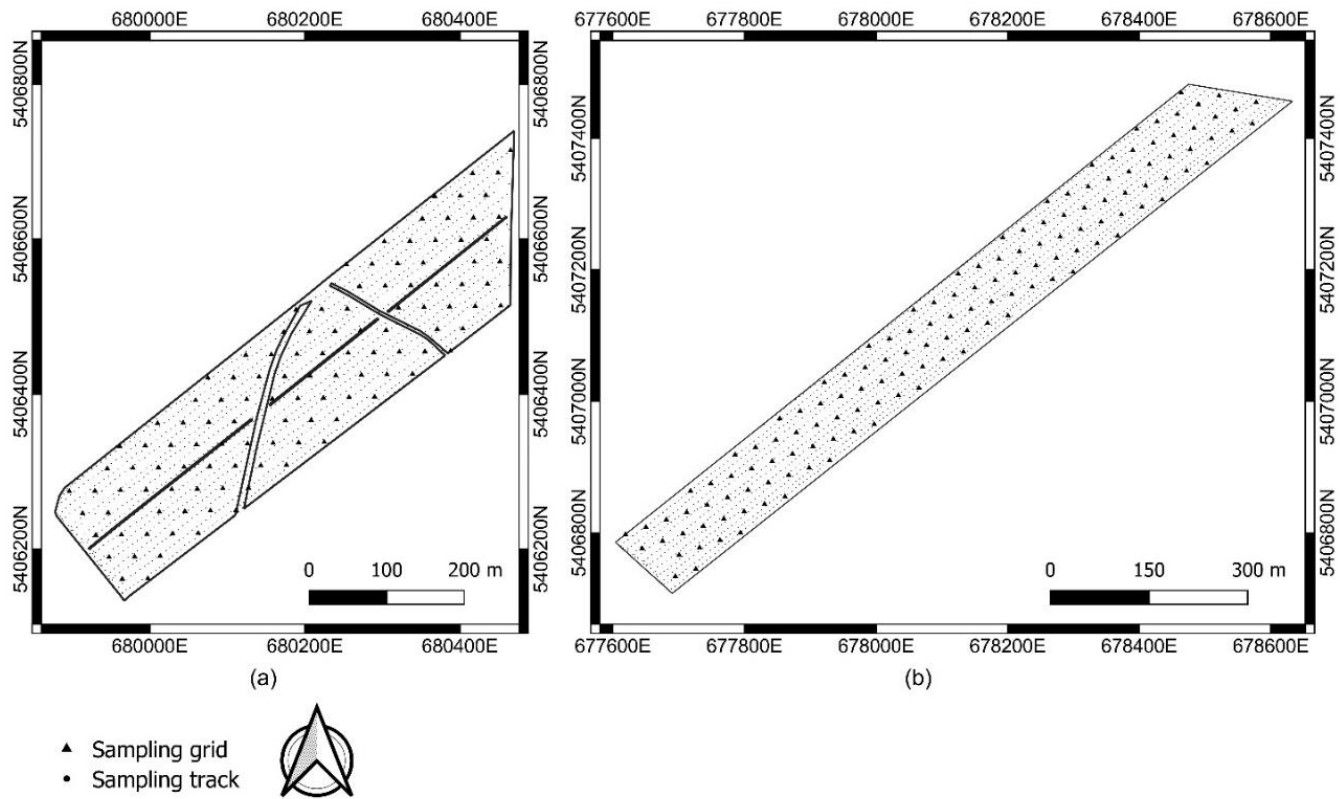

**Figure A1.** Sampling strategy and field layouts of (**a**) Field$_{Flat}$ and (**b**) Field$_{Und}$ of selected wild blueberry fields in Normandin, QC. The sensor track represents DUALEM EC$_a$ and RTK elevation sensor track.

**Table A1.** Summary of ratio-based VIs calculated from the SPOT image to capture variations in fruit yield density. The various spectral bands used in the equations are near-infrared (NIR) (760–890 nm), red (R) (625–695 nm), green (G) (530–590 nm), and blue (B) (450–520 nm).

| Name | Formula | Reference |
|---|:---:|:---:|
| Normalized Difference Vegetation Index (NDVI) | $\frac{NIR-R}{NIR+R}$ | [76] |
| Transformed Difference Vegetation Index (TDVI) | $TDVI = \sqrt{0.5 + \frac{NIR-R}{NIR+R}}$ | [77] |
| Optimized Soil Adjusted Vegetation Index (OSAVI) | $\frac{NIR-R}{NIR+R+0.16}$ | [78] |
| Non-Linear Index (NLI) | $\frac{NIR^2-R}{NIR^2+R}$ | [79] |
| Modified Simple Ratio (MSR) | $\frac{\left(\frac{NIR}{R}\right)-1}{\left(\sqrt{\frac{NIR}{R}}\right)+1}$ | [80] |
| Green Ratio Vegetation Index (GRVI) | $\frac{NIR}{G}$ | [81] |
| Green Difference Vegetation Index (GDVI) | $NIR - G$ | [82] |
| Enhanced Vegetation Index (EVI) | $2.5 * \frac{(NIR-R)}{(NIR+6*R-7.5*B+1)}$ | [83] |
| Modified Soil Adjusted Vegetation Index (MSAVI2) | $\frac{2*NIR+1-\sqrt{(2*NIR+1)^2-8(NIR-R)}}{2}$ | [84] |

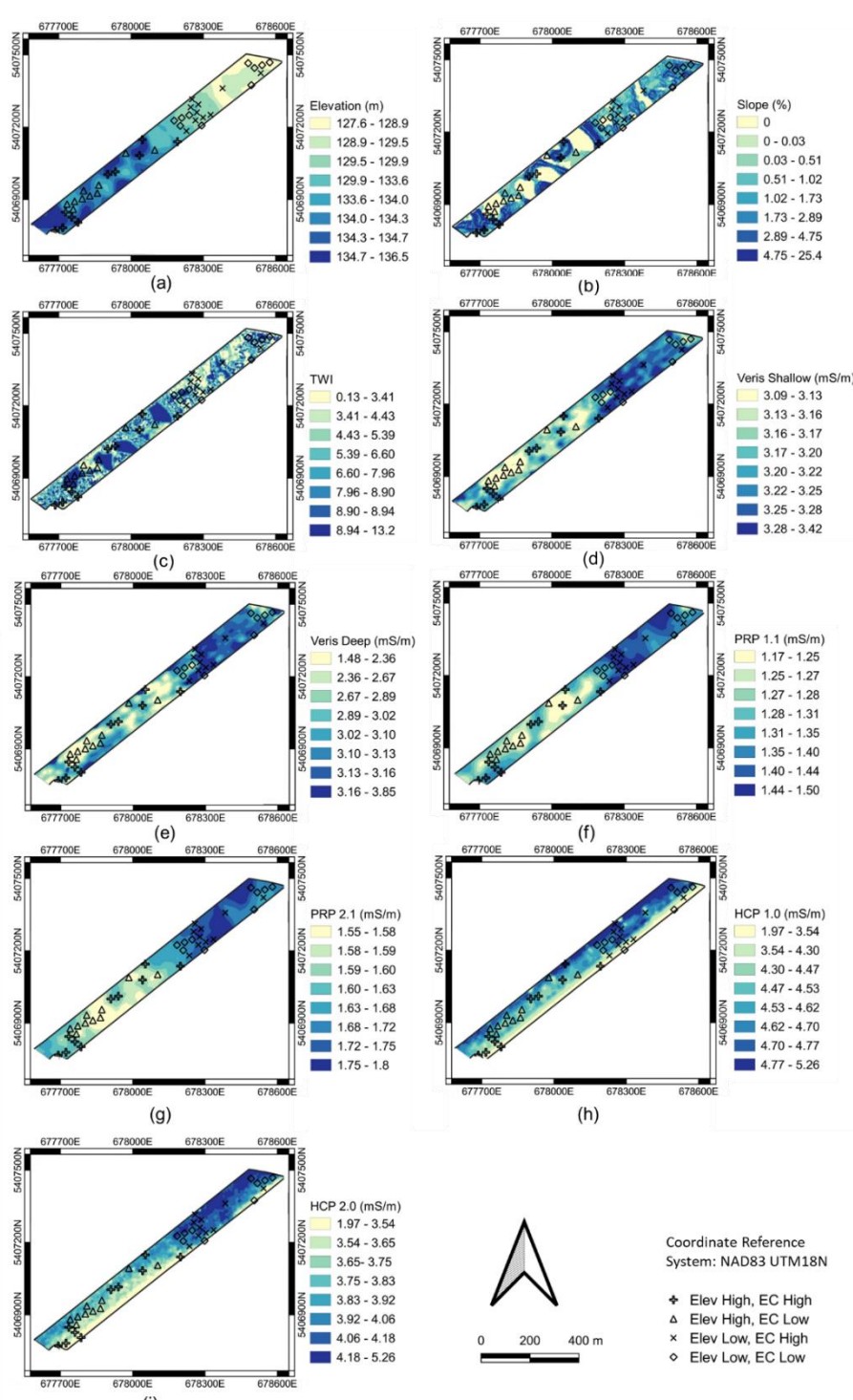

**Figure A2.** Ordinary kriging interpolated maps in study site Field$_{Und}$ of (**a**) elevation, (**b**) derived slope, (**c**) derived topographic wetness index (TWI), (**d**) Veris soil EC$_a$ shallow depth, (**e**) Veris soil EC$_a$ deep, (**f**) DUALEM PRP1.1, (**g**) PRP2.1, (**h**) HCP 1.0, and (**i**) HCP 2.0.

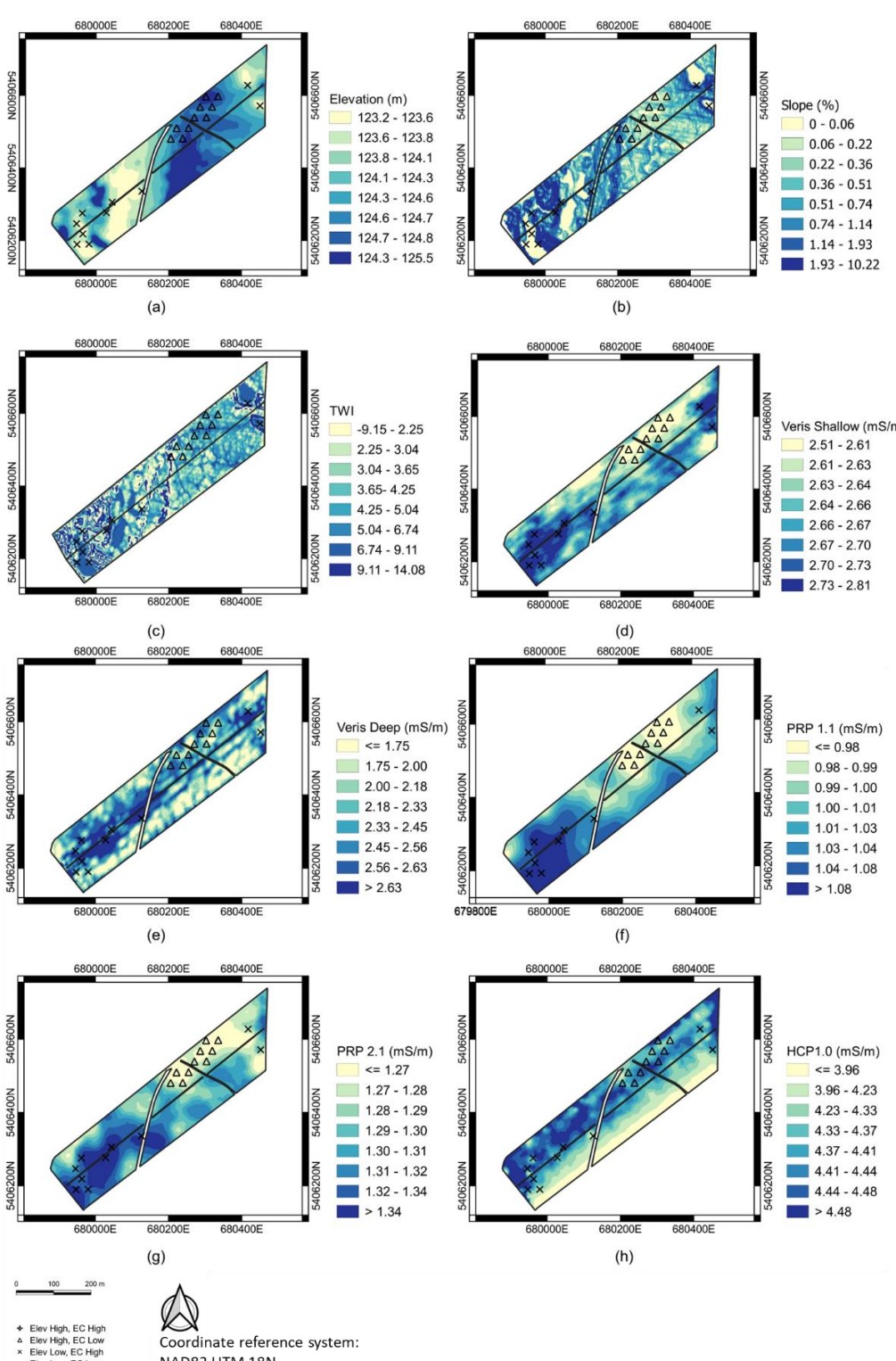

**Figure A3.** Ordinary kriging interpolated maps in study site Field$_{Flat}$ of (**a**) elevation, (**b**) derived slope, (**c**) derived topographic wetness index (TWI), (**d**) Veris soil EC$_a$ shallow depth, (**e**) Veris soil EC$_a$ deep, (**f**) DUALEM PRP1.1, (**g**) PRP2.1, (**h**) HCP1.0.

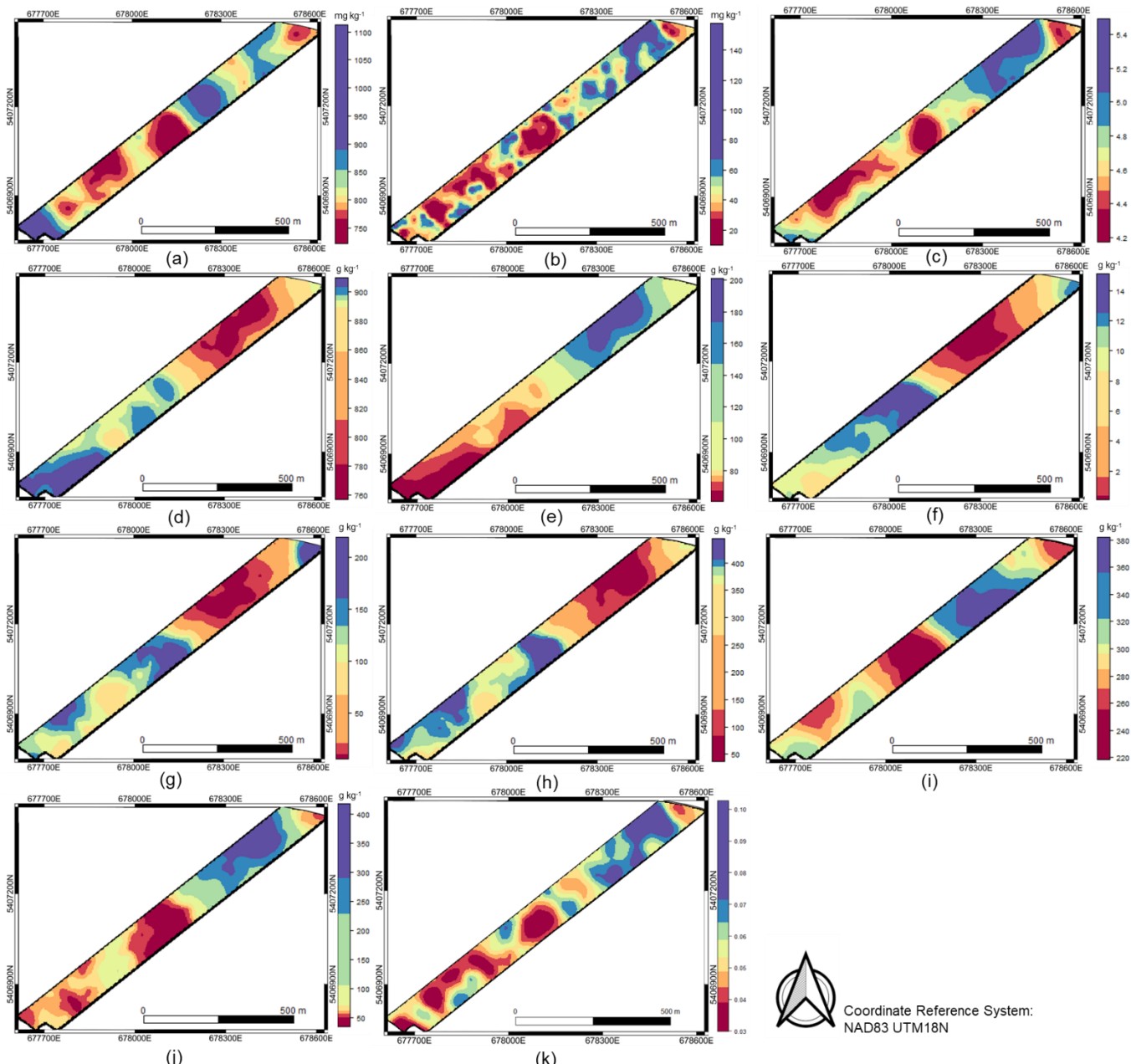

**Figure A4.** Kriging-interpolated maps in Field_Und of soil attributes (**a**) Al, (**b**) P, (**c**), pH, (**d**) total Sand, (**e**) total Silt, (**f**) Very coarse sand, (**g**) Coarse sand, (**h**) Medium sand, (**i**) Fine sand, (**j**) Very fine sand, (**k**) P:Al ratio.

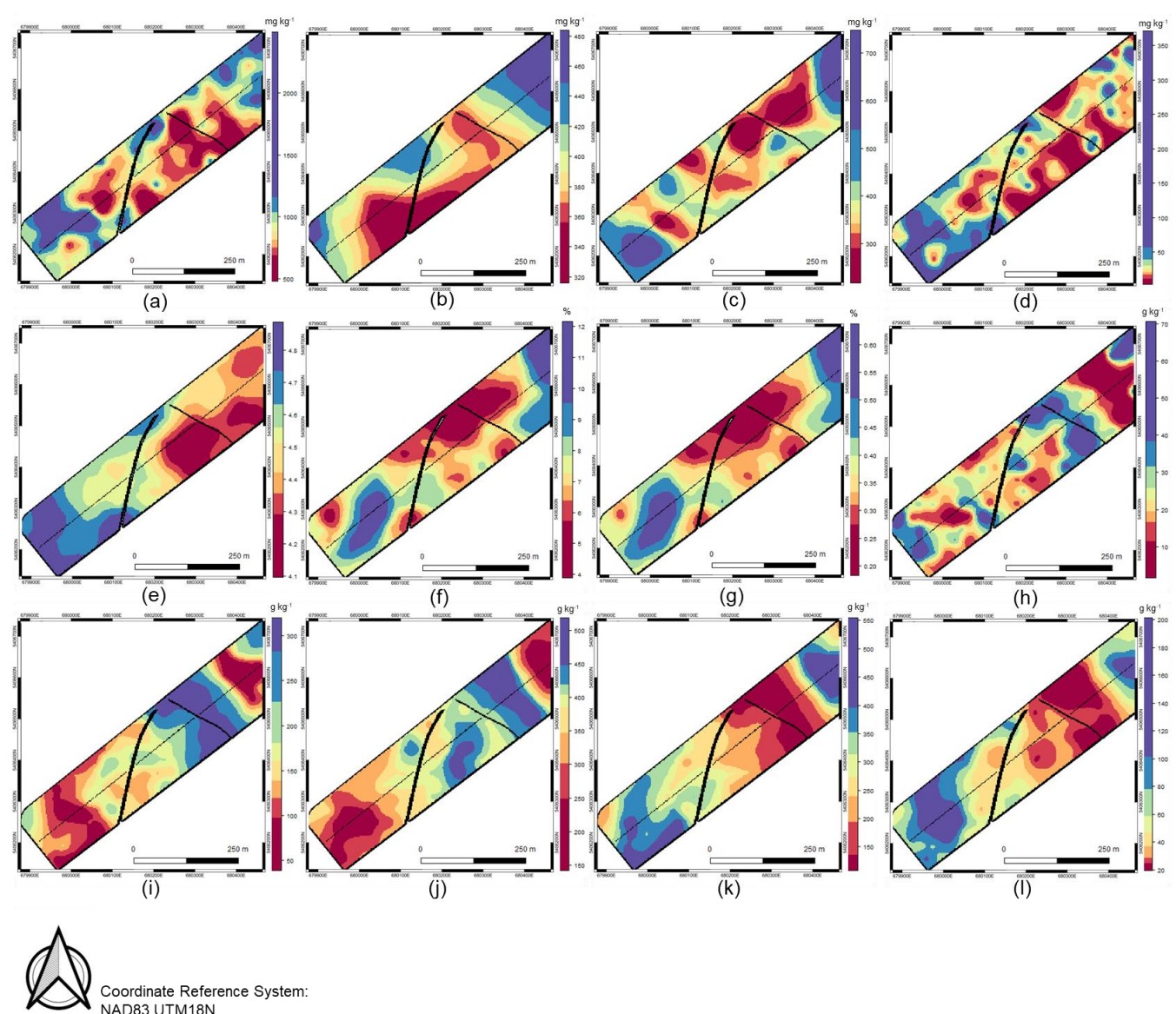

**Figure A5.** Kriging-interpolated maps in Field$_{Flat}$ of soil attributes (**a**) Al, (**b**) Ca, (**c**) Fe, (**d**) P, (**e**) pH, (**f**) total C, (**g**) total N, (**h**) Very coarse sand (**i**) Coarse sand (**j**) Medium sand, (**k**) Fine sand, (**l**) Very fine sand.

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
