# Peer review of "Proximal and Remote Sensing Data Integration to Assess Spatial Soil Heterogeneity in Wild Blueberry Fields"

_soilsystems, doi:10.3390/soilsystems6040089_

Round 1

Reviewer 1 Report

I have several suggestions and questions for authors as below.

1.      According to the main title and lines 82-83, I didn’t see the kind of site-specific nutrient management and variable rate fertilization in your manuscript. The “Discussion” section is a place to visualize the primary goal of your research.

2.       Line 65, what is the MZ?

3.       Is the second objective (lines 89-91) a part of the first objective or different according to your main topic of site-specific nutrient management?

4.       The flow chart is better for visualizing your research flow.

5.       All maps in the manuscript should have geographic coordinates (all maps) and symbology of all attributes (Figures A2 and A3).

6.       The brief explains how the proximal soil sensing work is needed. I suggest drawing the schematic overview of the sensors regarding the HCP and PERP.

7.       Lines 165-166, when should the radiometrically and atmospherically correction be before or after pan-sharpened?

8.       I think the PCA components are affected by the input of variables (bands), so I suggested testing all PCA components (PCA 1 til PCA 4, line 184) to identify/distinguish blueberry bushes, yield, and bare spots. The interpretation accuracy using satellite remote sensing data is needed.

9.       I suggest distinguishing a part for the “Method” section and the “ Results” section in lines 224-279.

Author Response

Reviewer 1

Comments and Suggestions for Authors

I have several suggestions and questions for authors as below.

  1. According to the main title and lines 82-83, I didn’t see the kind of site-specific nutrient management and variable rate fertilization in your manuscript. The “Discussion” section is a place to visualize the primary goal of your research.

The title has been clarified.

  1. Line 65, what is the MZ?

“Management zone”. The full name has been added the first time it is referenced.

  1. Is the second objective (lines 89-91) a part of the first objective or different according to your main topic of site-specific nutrient management?

The objectives have been clarified in the text.

  1. The flow chart is better for visualizing your research flow.

 Inserted into Material and Methods

  1. All maps in the manuscript should have geographic coordinates (all maps) and symbology of all attributes (Figures A2 and A3).

 Changed

  1. The brief explains how the proximal soil sensing work is needed. I suggest drawing the schematic overview of the sensors regarding the HCP and PERP.

A schematic overview was added to Materials and Methods section 2.4.

  1. Lines 165-166, when should the radiometrically and atmospherically correction be before or after pan-sharpened?

Pan-sharpening is done first. The reason was clarified in the text.  

  1. I think the PCA components are affected by the input of variables (bands), so I suggested testing all PCA components (PCA 1 til PCA 4, line 184) to identify/distinguish blueberry bushes, yield, and bare spots. The interpretation accuracy using satellite remote sensing data is needed.

Added to Results section 3.4.

  1. I suggest distinguishing a part for the “Method” section and the “ Results” section in lines 224-279.

Because certain preliminary results were used to make key decisions in the methodology (e.g., ECa correlation values used to determine which sensing depths to use for the theoretical combinations), we chose to present them in the methodology and only show results in the results section. A flowchart of the methodology was added to better illustrate how these preliminary results are integral to the methodology.

Reviewer 2 Report

The paper presents a contribution to the field of site-specific nutrient management of wild blueberry.

In general, the article is well-organized and contains all expected components, from introduction to conclusions. In general, all sections are well-developed and clearly explained. English is not my native language, but I feel that the article is well-written and easy to understand. The article supposes a major advance to the knowledge base. 

A very promising suggestion expecting to improve the clarity of the performed research and their results could go through:

1) replace PCA analysis with cluster analysis. This would allow obtaining better defined classes for the evaluated parameters (apparent soil electrical conductivity (ECa), topographic attributes, and multi-spectral satellite im-18 agery). The purpose of these analyzes isto establish functional groups of correlated experimental treatments. The R package 'Cluster' would be very suitable.

2) Then, the relationship between the classes of fruit yield and the mentioned parameters could be estimated using the most robustness linear and/or nonlinear machine learning algorithms. These algorithms could be fitted to the abovementioned parameters as explanatory variables, and the fruit yield class could be the response variable. The dataset should be partitioned x%–x% into training and test sets, and x-fold cross-validation should be used to estimate the accuracy of the models. The classifiers that learn the best should be then evaluated on the test set and used to identify the groups that were homogeneous in terms of fruit yield. As a suggestion, the machine learning algorithms could be simultaneously implemented using the Python software.

Author Response

Reviewer - 2

Comments and Suggestions for Authors

The paper presents a contribution to the field of site-specific nutrient management of wild blueberry.

In general, the article is well-organized and contains all expected components, from introduction to conclusions. In general, all sections are well-developed and clearly explained. English is not my native language, but I feel that the article is well-written and easy to understand. The article supposes a major advance to the knowledge base. 

A very promising suggestion expecting to improve the clarity of the performed research and their results could go through:

1) replace PCA analysis with cluster analysis. This would allow obtaining better defined classes for the evaluated parameters (apparent soil electrical conductivity (ECa), topographic attributes, and multi-spectral satellite imagery). The purpose of these analyzes is to establish functional groups of correlated experimental treatments. The R package 'Cluster' would be very suitable.

 PCA is involved in several clustering algorithms. Although it does not produce management zones, PCA allowed us to assess contribution of different variables to the ability to separate field areas with different agroclimatic conditions. This does not substitute clustering analysis but helps answer the principal question if clustering would be appropriate, considering the list of variables and the number of ground truth datapoints (degrees of freedom).

2) Then, the relationship between the classes of fruit yield and the mentioned parameters could be estimated using the most robustness linear and/or nonlinear machine learning algorithms. These algorithms could be fitted to the abovementioned parameters as explanatory variables, and the fruit yield class could be the response variable. The dataset should be partitioned x%–x% into training and test sets, and x-fold cross-validation should be used to estimate the accuracy of the models. The classifiers that learn the best should be then evaluated on the test set and used to identify the groups that were homogeneous in terms of fruit yield. As a suggestion, the machine learning algorithms could be simultaneously implemented using the Python software.

Normally, machine learning would require an increased number of point measurements. Our goal was to explore agronomic reasoning for yield variability instead of a black-box approach, which would probably overfit our rather limited dataset.

Reviewer 3 Report

The present paper describes the new prediction method of fruit yield of wild blueberry by in situ measurement of soil properties together with satellite image analysis. Although authors' investigation does not directly relate to the site-specific management of nutrients for blueberry cultivation, the paper provides useful information on the authors' final objective of site-specific management. Thus, the paper is acceptable after minor revision.

Line 82: The abbreviation 'SSM' appears first time in the main text. Authors should show the definition of 'SSM' here.

Lines 169–171: Authors should describe about the dataset for PCA. Did authors use dataset including both 'Flat' and 'Und'? Or did authors analyze separately?

Lines 205–210: Authors should describe the method of soil electrical conductivity in more detail including the characteristics of probes. Soil EC is usually affected by both soil moisture and salinity. If authors want to get information on soil moisture, why did authors never use TDR data directly? Authors should include more exact description the reason why authors selected EC as one variable showing soil characteristics.

Figure 1 and Figure 2: 'ms' on the vertical axis should be 'mS'.

Table 1: '3.18' in the 3rd line of foot note should be '3.18 m'.

Line 327: 'P' of 'Phosphorous' should be lower case letter.

Lines 414, 525, 529, 533–535: 'a' of 'ECa' should be subscript.

Line 646: 'FieldUnd' should be 'FieldFlat'.

Author Response

Reviewer 3

Comments and Suggestions for Authors

The present paper describes the new prediction method of fruit yield of wild blueberry by in situ measurement of soil properties together with satellite image analysis. Although authors' investigation does not directly relate to the site-specific management of nutrients for blueberry cultivation, the paper provides useful information on the authors' final objective of site-specific management. Thus, the paper is acceptable after minor revision.

Line 82: The abbreviation 'SSM' appears first time in the main text. Authors should show the definition of 'SSM' here.

Changed

Lines 169–171: Authors should describe about the dataset for PCA. Did authors use dataset including both 'Flat' and 'Und'? Or did authors analyze separately?

The satellite image containing both fields was pre-processed in whole, but each field was subset from the larger image before calculating the VIs or running PCA. This information has been clarified in the Methods section.

Lines 205–210: Authors should describe the method of soil electrical conductivity in more detail including the characteristics of probes. Soil EC is usually affected by both soil moisture and salinity. If authors want to get information on soil moisture, why did authors never use TDR data directly? Authors should include more exact description the reason why authors selected EC as one variable showing soil characteristics.

TDR is a point-based solution while the aim of this study was to examine proximal ECa sensing. This has been clarified in the introduction.

Figure 1 and Figure 2: 'ms' on the vertical axis should be 'mS'.

 Changed

Table 1: '3.18' in the 3rd line of foot note should be '3.18 m'.

 Changed in all tables.

Line 327: 'P' of 'Phosphorous' should be lower case letter.

 Changed

Lines 414, 525, 529, 533–535: 'a' of 'ECa' should be subscript.

 Changed

Line 646: 'FieldUnd' should be 'FieldFlat'.

Changed